# 3D Visual Illusion Depth Estimation

**Chengtang Yao**[1,2][†] **Zhidan Liu**[1,2][†] **Jiaxi Zeng**[1,2], **Lidong Yu**[3,4], **Yuwei Wu**[1,2][*] **Yunde Jia**[2,1][*]

[1]Beijing Key Laboratory of Intelligent Information Technology,
School of Computer Science & Technology, Beijing Institute of Technology, China
[2]Guangdong Provincial Key Laboratory of Machine Perception and Intelligent Computing,
Shenzhen MSU-BIT University, Shenzhen, China
[3]NVIDIA, [4]NEOLIX
{zdliu, wuyuwei, jiayunde}@bit.edu.cn
{yao.c.t.adam, yvlidong, jiaxizeng.jx}@gmail.com

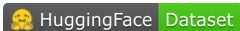 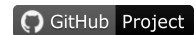 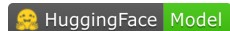

## Abstract

3D visual illusion is a perceptual phenomenon where a two-dimensional plane is manipulated to simulate three-dimensional spatial relationships, making a flat artwork or object look three-dimensional in the human visual system. In this paper, we reveal that the machine visual system is also seriously fooled by 3D visual illusions, including monocular and binocular depth estimation. In order to explore and analyze the impact of 3D visual illusion on depth estimation, we collect a large dataset containing almost 3k scenes and 200k images to train and evaluate SOTA monocular and binocular depth estimation methods. We also propose a 3D visual illusion depth estimation framework that uses common sense from the vision language model to adaptively fuse depth from binocular disparity and monocular depth. Experiments show that SOTA monocular, binocular, and multi-view depth estimation approaches are all fooled by various 3D visual illusions, while our method achieves SOTA performance.

## 1 Introduction

Depth estimation aims to recover the 3D geometry of a scene from a single image or an image sequence. It is a long-standing and challenging vision problem, with extensive research in monocular depth estimation [46, 47, 3, 17], stereo matching [27, 42, 5, 13], and multi-view reconstruction [41, 39]. These works have achieved impressive performance in typical, well-structured scenes, approaching human-level perception. However, beyond typical scenes, there exist many 3D visual illusion scenes that make a flat artwork or object look three-dimensional, as illustrated in Figure 1. These 3D visual illusions mislead the depth perception and seriously affect the downstream applications, causing safety-critical risks in AR/VR and robotics.

In this paper, we present a 3D-Visual-Illusion dataset to investigate the impact of 3D visual illusions on depth estimation. The dataset includes five types of illusions: inpainting illusion (e.g., inpainting on walls or floors), picture illusion (e.g., image printed/drawn on a paper), replay illusion (e.g., videos replayed on different screens), holography illusion, and mirror illusion (e.g., specular and transparent surfaces). It comprises nearly 3,000 scenes and 200,000 images, covering various environments from small objects to large scenes and from indoor to outdoor settings. We construct the dataset from both virtual and real-world data. Virtual data is generated using two separate pipelines: one based on web-sourced videos and the other on generative models. Real-world data is captured using a stereo camera and a solid-state LiDAR depth sensor.

---

[†]These authors contributed equally to this work.
[*]Corresponding author.

39th Conference on Neural Information Processing Systems (NeurIPS 2025).

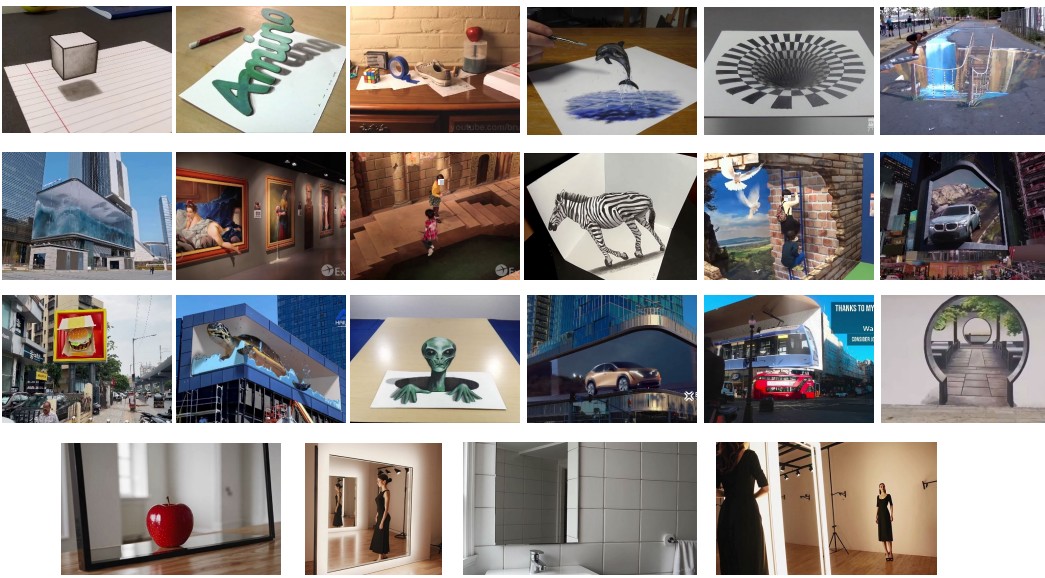

Figure 1: The visualization of 3D visual illusions.

The evaluation results on the 3D-Visual-Illusion dataset reveal distinct failure modes for different SOTA depth estimation models. Monocular methods, which rely on the mapping from texture cues to 3D geometry, are easily misled by illusion patterns such as printed images or screen content. In contrast, stereo methods depend on pixel correspondences and fail on transparent or reflective surfaces like glass and mirrors, where conflicting signals distort the matching process. Notably, stereo and monocular methods exhibit complementary strengths, often succeeding where the other fails. Stereo methods succeed on texture-rich illusions, while monocular models can recover mirror geometry through learned priors. Yet, each alone is insufficient to handle the full spectrum of 3D visual illusions, and this complementarity motivates us to seek a unified framework that leverages the strengths of both.

Inspired by the strong generalization capability of vision-language models (VLMs) on mirror illusions (see the supplementary materials for details), we propose a VLM-driven monocular–stereo fusion framework to fuse stereo and monocular priors. The model leverages commonsense knowledge from VLMs to assess the reliability of monocular and binocular depth across different regions, enabling more effective depth fusion. Our model consists of two components: a dual-branch prediction network and a VLM-based fusion network. The dual-branch network takes a rectified image pair as input and simultaneously predicts monocular depth and binocular disparity. The VLM-based fusion network employs a pre-trained vision model to extract features from the left RGB image. These features are mapped into a shared embedding space using a large language model conditioned on a language prompt. The embedding features are then used to generate a confidence map via flow matching. The confidence map is used to align the affine-invariant monocular depth to metric scale, which is then fused with the binocular disparity to produce the final depth map. Experiments on our dataset and the Booster dataset demonstrate that our method achieves SOTA performance under a wide range of 3D visual illusions.

## 2 Related Work

### 2.1 Stere Matching

Stereo matching is a pixel-wise labeling task that relies on dense correspondence between a pair of images. The SOTA methods are either GRU-based iterative methods or Transformer-based methods. The former methods predict the disparity update and iteratively approximate the GT value in a GRU framework [27, 24, 16, 48, 5, 10]. They have achieved great performance in both benchmark and zero-shot generalization testing. The latter methods use a Transformer to learn matching and predict the disparity map [25, 12, 43, 44, 41, 23, 39]. The Transformer-based methods achieve superior performance by learning from large-scale data. In this paper, we collect a comprehensive, large-scale dataset to thoroughly investigate and evaluate the impact of 3D visual illusions on matching methods.

We reveal that stereo matching methods are highly susceptible to various illusions. Our method leverages common sense from VLM to detect mirror illusions and help rectify these illusions.

## 2.2 Monocular Depth Estimation

Monocular depth estimation is a pixel-wise regression task based on a single image. Recent deep learning methods leverage diffusion models [17, 8, 50] or Transformers [33, 2, 47, 45] to extract depth-related features, in both supervised and self-supervised settings [9, 11, 19]. Despite their impressive generalization performance across diverse scenes, these methods fundamentally rely on monocular cues, which are susceptible to 3D visual illusions, much like the human visual system. In this work, we introduce a large-scale benchmark to evaluate state-of-the-art monocular depth models under such illusions. Our results show that existing methods are consistently misled by these challenging patterns. To address this, we propose leveraging matching-based depth cues as complementary information to enhance monocular depth estimation.

## 2.3 Large Vision-Language Model

The large vision-language model (VLM) injects common sense from billions of textual data to support vision understanding and generation [49]. It presents great power in various tasks, like visual question answering, image generation, and navigation. The methods of these tasks mainly adapt a pre-trained VLM to specific datasets to preserve the generalization ability, while promoting the understanding of specific tasks. To further facilitate the training of VLM in downstream tasks, a lot of methods explore different finetuning strategies, like Prompt [4], Adapter [14], LoRA [15], and LST [35]. In this paper, inspired by the strong detection ability of VLM on mirror illusions, we use VLM to predict the confidence of the disparity map to recover the metric version of monocular depth. The common sense from large VLM is beneficial for the confidence estimation in various complex scenes.

# 3 3D Visual Illusion Dataset

We construct the 3D-Visual-Illusion dataset to investigate the challenges posed by 3D visual illusions in depth estimation. The dataset comprises nearly 3,000 scenarios, with over 200,000 frames for training and 617 frames for testing. It includes images of various resolutions, up to a maximum of $1080 \times 1920$, and spans a wide range of scenes, from indoor environments and small objects to large-scale street views. The dataset covers five types of illusions: inpainting illusion (e.g., inpainting on a wall/floor), picture illusion (e.g., picture printed/drawn on a paper), replay illusion (e.g., video replayed on a different screen), holography illusion, and mirror illusion (e.g., specular or transparent surfaces). Data is collected from both virtual and real-world sources. Details of the construction process for the virtual and real subsets are provided in the following sections.

## 3.1 Virtual Data

We collect a large amount of video data from websites and text-to-video generative models. We take the videos as left image sequences and generate disparity maps and right images.

**Video Collection** We adopt two distinct data collection strategies for the first four types of illusions and for mirror illusions. For inpainting, picture, replay, and holography illusions, we crawl 5,226 web videos (over 52M frames) using keyword-based search. We then apply a vision-language model, Qwen2-VL-72B [1, 40], to automatically filter out irrelevant frames, reducing the dataset to 4,519 videos (1.4M frames). Further manual filtering removes blurry or occluded frames, resulting in 1,384 high-quality videos with 236K frames. Mirror illusions are difficult to collect from the web due to the rarity of mirror-related keywords and high-quality videos. To address this, we generate videos using SOTA generative models, including Sora [30], Kling [21], and HunyuanVideo [20]. Prompts are initially created with ChatGPT and manually refined. Videos violating physical plausibility are discarded. In total, we collect 234 high-quality mirror illusion videos comprising 2,382 frames.

**Depth Generation** After collecting videos from both web sources and generative models, we generate depth for each frame using the pipelines illustrated in Figure 2 and 3. Different pipelines are used for the two data sources for the following reasons: (1) Web-sourced videos typically involve fixed cameras, providing limited viewpoints and making accurate scene reconstruction difficult. (2) Generative videos are used primarily for mirror illusions, which require modeling the geometry of the

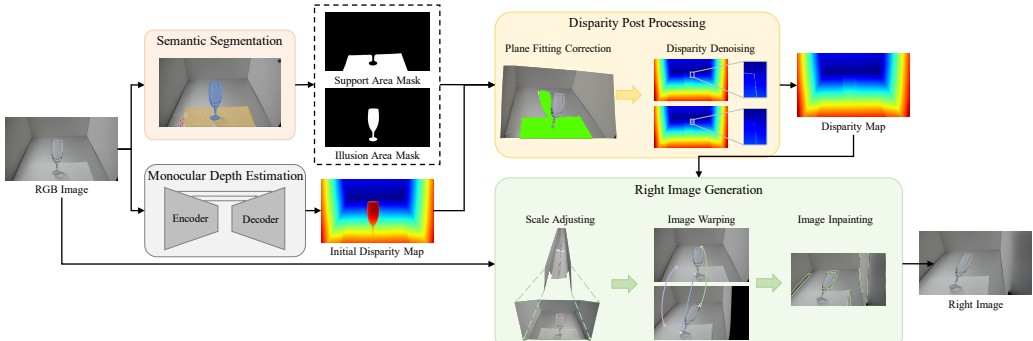

Figure 2: The data generation pipeline for web-sourced data.

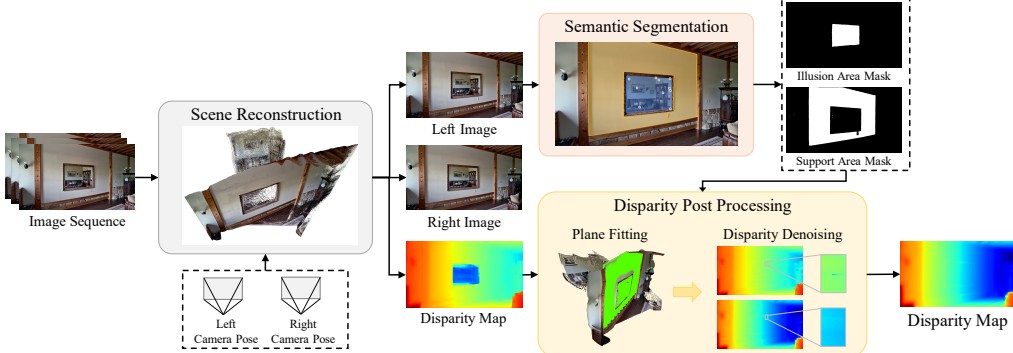

Figure 3: The data generation pipeline for videos from generative models.

reflected (mirror) world to generate right-view images. However, monocular depth estimation may ambiguously predict either the mirror surface or the reflected scene, leading to inconsistent results.

For web videos, we use the pre-trained DepthAnything V2 [46] to predict inverse depth, which is treated as disparity under unknown camera parameters. However, in regions affected by 3D visual illusions, the predicted disparity is often severely inaccurate. To correct this, we introduce a neighboring support region, assuming it lies on the same plane as the illusion region, and use it as a reference for disparity correction. Segmentation masks for both illusion and support regions are obtained using SAM2 [34]. Since the automatic mode struggles to detect illusions accurately, we manually annotate all frames using SAM2's click mode, removing redundant or imperceptible cases.

After obtaining the mask of illusion and support regions, we fit a plane using the points within a support region. In the standard 3D camera coordinate system $(X, Y, Z)$, the general form of a plane with parameters $(\alpha, \beta, \gamma, \delta)$ is $\alpha \cdot x + \beta \cdot y + \gamma \cdot z + \delta = 0$. Given the relationship between image coordinate $(u, v)$, disparity $d$, and $(X, Y, Z)$:

$$(u, v, d) = \frac{1}{z}(x, y, B) \cdot (f_x, f_y, \frac{f_x + f_y}{2}) + (c_x, c_y, 0),$$

the planar structures in 3D space $(X, Y, Z)$ remain planar in the disparity space $(u, v, d)$ with plane parameters $(\alpha, \beta, \delta, \gamma)$: $\alpha \cdot u + \beta \cdot v + \delta \cdot d + \gamma = 0$. This property is crucial for our generation, as it allows plane fitting directly in disparity space under unknown camera intrinsics and baseline, avoiding the need to convert disparity into depth. Given a set of $N$ points from the support region, $\{(u_i, v_i, d_i)\}_{i=1}^N$, the goal of plane fitting becomes a least squares fitting problem over parameters $(\alpha, \beta, \delta, \gamma)$. To mitigate the impact of noise during fitting, we adopt RANSAC [7] for robust plane estimation (see supplemental materials for details). The fitted plane is then used to rectify disparity values within the illusion regions. After obtaining the rectified disparity map, we apply an additional denoising step to ensure smooth transitions along the boundaries between support regions and their surroundings.

For videos from generative models, we reconstruct the entire scene using InstantSplat [6], which first estimates geometry via DUSt3R [41] and synthesizes novel views using Gaussian Splatting (GS) [18].

We extract the disparity map from the geometry derived from DUSt3R and refine it through a series of post-processing steps, including semantic segmentation, RANSAC-based plane fitting, and disparity denoising.

**Right Image Generation**    The right-view images for generative-model videos are directly rendered using Gaussian Splatting (GS). For web-sourced videos, right views are generated by warping the left images using monocular disparity. Due to scale ambiguity, we estimate an optimal scale factor $s$ via binary search, terminating when most warped pixels fall within the valid image width:

$$\tilde{s} = \arg\min_s \left| \frac{\sum_x \mathbb{1}(0 \leq u - s \cdot d_{(u,v)} < W)}{N} - \tau \right|, \tag{1}$$

where $\mathbb{1}(\cdot)$ is the indicator function, $d$ is the disparity, $(u, v)$ are pixel coordinates, $W$ is the image width, $N$ is the number of pixels, and $\tau$ is the target ratio of valid pixels. We use the scaled disparity $\tilde{d} = \tilde{s} \cdot d$ to warp the left image accordingly. In cases where multiple source pixels are warped to the same target location due to occlusions, we retain the one with the largest disparity to maintain consistency:

$$I_r(u', v) = I_l(u^*, v),$$
$$u^* = \arg\max_u \{d_{(u,v)} \mid u - \tilde{d}_{(u,v)} = u'\}. \tag{2}$$

To address holes after warping, we apply an image inpainting method [36] to produce visually complete right-view images. The full generation algorithm is detailed in the supplemental materials.

## 3.2   Real-world Data

In addition to virtual data, we collect real-world data comprising 72 scenes and 617 frames. The setup includes a stereo camera (ZED Mini) and a LiDAR-based depth sensor (Realsense L515), with details provided in the supplemental materials. To ensure accurate alignment, the two sensors are rigidly mounted, and their relative pose is calibrated using a checkerboard. The L515 depth map is then warped to the ZED left camera frame based on the calibration to construct the ground-truth depth.

Due to the lower resolution of the L515, direct pixel-wise warping to the higher-resolution ZED image will result in sparse depth maps and incorrect projections, particularly in occluded regions where background depths may overwrite foreground pixels. To address this, we first densify the L515 point cloud by upsampling its depth map $\mathcal{Z}_L$ via nearest-neighbor interpolation and proportionally scaling its intrinsic matrix $K_L$.

After densifying the point cloud, image coordinates from the L515, $(U_L, V_L)$, are first projected into the 3D camera coordinate $(X_L, Y_L, Z_L)$ using the L515 depth map. These 3D points are then transformed to the ZED left camera's coordinate system $(X_Z, Y_Z, Z_Z)$. Finally, they are projected onto the ZED left image coordinates $(U_Z, V_Z)$:

$$[X_L, Y_L, Z_L] = Z_L \cdot K_L^{-1} \cdot [U_L, V_L, 1],$$
$$[X_Z, Y_Z, Z_Z] = R \cdot [X_L, Y_L, Z_L] + T, \tag{3}$$
$$[U_Z, V_Z, 1] = K_Z \cdot [X_Z/Z_Z, Y_Z/Z_Z, 1].$$

Here, $R \in \mathbb{R}^{3 \times 3}$ and $T \in \mathbb{R}^{3 \times 1}$ denote the rotation and translation matrices between the L515 and the ZED cameras, both obtained via calibration. $K_L \in \mathbb{R}^{3 \times 3}$ and $K_Z \in \mathbb{R}^{3 \times 3}$ represent the intrinsic matrices of the L515 and the ZED left camera, respectively.

In the projected coordinates $(U_Z, V_Z)$, multiple 3D points $P_m$ may map to the same pixel due to slanted surfaces or occlusions. To resolve this, we apply Z-buffering to retain the point with the minimum depth for the ZED depth map $\mathcal{Z}_Z$:

$$\mathcal{Z}_Z(u_Z, v_Z) = z_Z^*,$$
$$z_Z^* = \min_{(u_Z', v_Z', z_Z') \in P_m} \{Z_Z' \mid (u_Z', v_Z') = (u_Z, v_Z)\}. \tag{4}$$

Although upsampling greatly densifies the point cloud, projecting from a lower-resolution to a higher-resolution space may still introduce small holes. To address this, we apply connected component

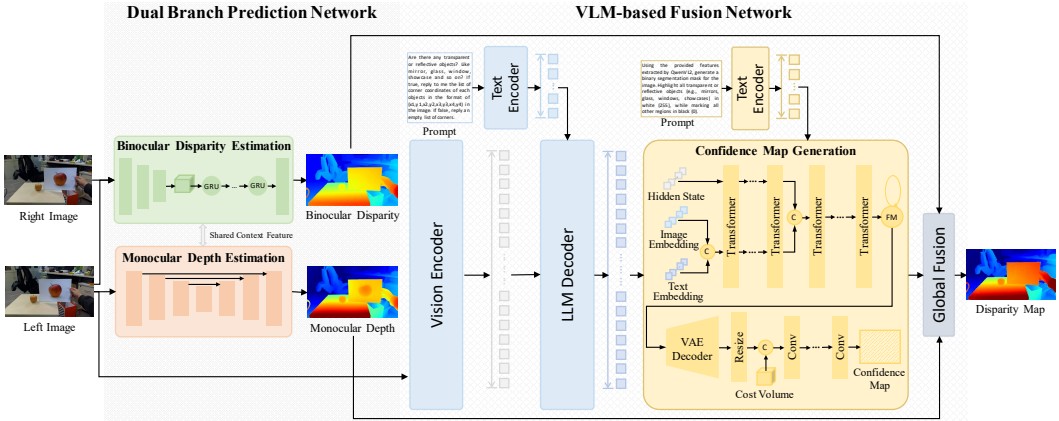

Figure 4: The pipeline of VLM-driven binocular and monocular disparity fusion model.

analysis to identify missing regions and fill them using image inpainting [38]. The inpainted depth values are further smoothed to ensure seamless transitions with surrounding areas.

Furthermore, to identify unreliable depth values, the depth map projected onto the ZED image is reprojected back to the L515 coordinate:

$$
\begin{gathered}
[X_{Z\to L}, Y_{Z\to L}, Z_{Z\to L}] = R^{-1} \cdot (Z_Z \cdot K_Z^{-1} \cdot [U_Z, V_Z, 1] - T), \\
[U_{Z\to L}, V_{Z\to L}, 1] = \frac{1}{Z_{Z\to L}} \cdot K_L[X_{Z\to L}, Y_{Z\to L}, Z_{Z\to L}].
\end{gathered}
\tag{5}
$$

The reprojected pixels that correspond to invalid depth values or exhibit large depth differences with their ZED counterparts are marked as unreliable:

$$
\mathcal{Z}_Z(u_Z, v_Z) = \begin{cases} 0 & \text{if } \mathcal{Z}_L(u_{Z\to L}, v_{Z\to L}) == 0 \text{ or} \\ & |z_{Z\to L} - \mathcal{Z}_L(u_{Z\to L}, v_{Z\to L})| > \epsilon, \\ \mathcal{Z}_z(u_z, v_z) & \text{otherwise.} \end{cases}
\tag{6}
$$

Here, $\epsilon$ is a manually defined threshold. To further refine depth quality, median filtering is applied to suppress noise and remove outlier points from the point cloud. Finally, the depth map $\mathcal{Z}_Z$ is converted into disparity map $D$:

$$
D = B \cdot F / \mathcal{Z}_Z,
\tag{7}
$$

where $B$ denotes the baseline between the ZED's stereo cameras, and $F$ is the focal length of the ZED camera. The entire algorithm is detailed in the supplemental materials.

## 4 VLM-Driven Monocular-Stereo Fusion Model

As illustrated in Figure 4, our model first uses a dual-branch prediction network to predict the binocular disparity and the monocular depth. Then, a VLM-based fusion network is used to produce the final disparity map by fusing the binocular disparity and the monocular depth.

### 4.1 Dual-Branch Prediction Network

The dual-branch prediction network comprises a binocular disparity estimation branch and a monocular depth estimation branch. The stereo branch adopts an iterative optimization framework, extracting features from rectified image pairs and constructing a cost volume via dot product. Starting from an initial disparity of zero, a GRU-based module iteratively refines the disparity map. The monocular branch takes the left image as input and uses the frozen DepthAnything V2 [46] to predict affine-invariant inverse depth, which is disparity under unknown camera parameters. We also extract features from frozen DepthAnything V2, followed by learnable convolutions. These adapted features serve as left-view context to guide disparity refinement in the stereo branch.

### 4.2 VLM-Based Fusion Network

As shown in Figure 4, the VLM-based fusion network contains three main stages: the VLM prediction stage, the confidence map generation stage, and the global fusion stage.

**VLM Prediction Stage**    In this stage, we utilize the pretrained QwenVL2-7B model [40, 1]. The visual prompt comprises the left image, binocular disparity map, and monocular disparity map. Since the textures that mislead monocular depth estimation are often too complex and diverse to be explicitly described, the language prompt is instead designed from the perspective of materials that typically confuse stereo matching. By leveraging the general reasoning capabilities of the vision-language model, we extract embedding features that help assess the relative reliability of monocular and binocular depth cues under language guidance.

**Confidence Map Generation Stage**    This stage aims to transform the embedding features back into image space to generate a confidence map. Inspired by the flow-matching framework Flux [26, 31], we learn a guided path flow from Gaussian noise to a complex confidence distribution:

$$y_{k_{i+1}} = y_{k_i} + \Delta K \cdot v_{k_i}(y_{k_i}, c_e), \tag{8}$$

where $k$ is uniformly sampled in the interval $[0, 1]$, $\Delta K = 1/K$, and $K$ is the total number of steps. $y_0$ is sampled from a prior Gaussian distribution, $y_{k_i}$ is the intermediate state at $i$-th sampling, $c_e$ is the conditional embedding, and $v_{t_k}$ denotes the predicted velocity field at time $t_k$ under conditions $(y_{t_k}, c_e)$.

Specifically, the language prompts are first mapped into the embedding space. The image and text embeddings are concatenated to form $c_e$, which is combined with the intermediate state $y_{t_k}$ and passed through a stack of Transformers. Cross-attention is used to inject conditional information, and the velocity field $v_{t_k}$ is predicted. After multiple iterations via Equation 8, the final state $y_1$ is reshaped into a 2D format and decoded into image space using a variational autoencoder. The resulting feature is then concatenated with the cost volume and passed through convolution layers to predict the confidence map $I_c$:

$$I_c = \sigma(\mathcal{F}_c([\mathcal{G}_c(y_1'), V(D_s)])), \tag{9}$$

where $\sigma$ is the sigmoid function, $\mathcal{F}_c$ denotes convolution, $\mathcal{G}_c$ is the VAE decoder, $y_1'$ is the reshaped $y_1$, and $V(D_s)$ is the cost volume sampled around binocular disparity $D_s$.

**Global Fusion Stage**    Global fusion first aligns monocular disparity $D_M$ to the absolute/metric disparity space and then fuses the aligned monocular disparity $\tilde{D}_m$ with binocular disparity $D_s$. The alignment is achieved by affine transformation parameters $s_m, t_m$:

$$\tilde{D}_m = s_m \cdot D_m + t_m,$$
$$s_m, t_m = \arg \min_{s_m, t_m} \sum_{(u,v)} (s_m \cdot D_m(u, v) + t_m - D_s(u, v))^2. \tag{10}$$

$s_m$ and $t_m$ are learned through convolutions on the concatenation of the monocular disparity $D_m$ and the binocular disparity $D_s$. Since $s_m$ and $t_m$ on low-confident regions are unreliable, we further refine the parameters by pooling $s_m$ and $t_m$ of the high-confident neighbors. After acquiring refined parameters $s_m$ and $t_m$, we compute the aligned monocular disparity $\tilde{D}_m$ using Equation 10. $\tilde{D}_m$, $D_s$, and $I_c$ are then concatenated and passed through convolutions and upsampling layers to generate the final high-resolution disparity map.

## 5    Experiment

We first pre-train the models on the SceneFlow dataset [28], and then fine-tune them on the virtual 3D-Visual-Illusion data. The fine-tuned models are evaluated on both the real-world 3D-Visual-Illusion data and the Booster training set [32]. We compare our method with monocular depth estimation approaches (DepthAnything V2 [46], Marigold [17], Metric3D [47], and DepthPro [3]), multi-view foundation models (Dust3R [41] and VGGT [39]), and stereo matching methods (RAFT-Stereo [27], Selective-IGEV [42], and MochaStereo [5]). For additional details on training procedures, loss functions, evaluation metrics, and prompt designs, please refer to the supplementary material.

### 5.1    Influence of 3D Visual Illusions

Table 1 presents a comprehensive comparison of the real-world data in the 3D-Visual-Illusion dataset. The real-world data is mainly composed of inpainting, picture, and replay illusions. The results of all compared methods are obtained from official code and weights, where the stereo methods use model weights pretrained on the SceneFlow dataset.

Table 1: Evaluation results on illusion regions for real-world data of the 3D-Visual-Illusion dataset. *align* means alignment using globally shared affine parameters computed from ground truth.

| Method | Finetune | Disparity Space | | | | Depth Space | | |
|---|---|---|---|---|---|---|---|---|
| | | EPE ↓ | bad2 ↓ | bad3 ↓ | bad5 ↓ | AbsRel ↓ | RMSE ↓ | $\delta_1$ ↑ |
| DA V2 [46] | × | 5.81 | 61.45 | 43.18 | 30.57 | 0.14 | 0.15 | 92.86 |
| Metric3D [47] | × | 12.46 | 94.11 | 91.14 | 82.05 | 0.34 | 0.29 | 48.97 |
| DA V2 metric [46] | × | 16.24 | 92.53 | 87.43 | 75.25 | 0.52 | 0.39 | 48.75 |
| DepthPro [3] | × | 12.26 | 87.08 | 80.60 | 62.43 | 0.28 | 0.25 | 65.92 |
| Marigold [17] | × | 21.16 | 65.67 | 59.67 | 53.19 | 0.45 | 0.37 | 63.65 |
| DA V2 metric [46] + align | × | 5.23 | 56.82 | 45.50 | 28.89 | 0.17 | 0.15 | 93.70 |
| Metric3D [47] + align | × | 5.70 | 66.26 | 50.92 | 40.43 | 0.17 | 0.17 | 94.80 |
| DepthPro [3] + align | × | 4.36 | 44.98 | 34.98 | 24.70 | 0.09 | 0.10 | 93.83 |
| Dust3R [41] | × | 6.74 | 52.89 | 45.31 | 36.61 | 0.25 | 0.22 | 87.09 |
| VGGT [39] | × | 6.16 | 53.32 | 44.89 | 37.20 | 0.13 | 0.12 | 78.46 |
| RAFT-Stereo [27] | × | 1.62 | 24.32 | 13.20 | 2.97 | 0.04 | 0.06 | 99.18 |
| Selective-RAFT [42] | × | 1.58 | 23.46 | 12.65 | 2.57 | 0.03 | 0.07 | 99.60 |
| Selective-IGEV [42] | × | 1.67 | 24.06 | 13.11 | 2.99 | 0.04 | 0.10 | 99.26 |
| MochaStereo [5] | × | 1.75 | 25.49 | 14.11 | 3.54 | 0.04 | 0.11 | 98.76 |
| StereoAnything [13] | × | 2.41 | 29.00 | 16.15 | 6.54 | 0.11 | 0.32 | 96.23 |
| ours | ✓ | 1.77 | 26.72 | 15.73 | 3.60 | 0.03 | 0.08 | 99.60 |

Table 2: Zero-shot generalization on the balanced set of Booster dataset with quarter resolution. *All*: all regions, *Trans*: transparent regions, *NonTrans*: nontransparent regions.

| Method | Finetune | All | | | | Trans | | | | NonTrans | | | |
|---|---|---|---|---|---|---|---|---|---|---|---|---|---|
| | | EPE ↓ | bad2 ↓ | bad3 ↓ | bad5 ↓ | EPE ↓ | bad2 ↓ | bad3 ↓ | bad5 ↓ | EPE ↓ | bad2 ↓ | bad3 ↓ | bad5 ↓ |
| DA V2 [46] | × | 3.16 | 48.83 | 36.98 | 21.22 | 7.19 | 77.98 | 69.02 | 50.91 | 2.91 | 47.25 | 35.09 | 19.25 |
| Metric3D [47] | × | 35.55 | 99.70 | 99.32 | 97.73 | 41.55 | 99.37 | 98.93 | 97.91 | 34.89 | 99.71 | 99.32 | 97.59 |
| DA V2 metric [46] | × | 21.55 | 94.28 | 91.37 | 84.21 | 28.42 | 93.04 | 90.72 | 86.78 | 20.94 | 94.25 | 91.22 | 83.84 |
| DepthPro [3] | × | 24.44 | 92.98 | 90.23 | 84.25 | 25.65 | 92.98 | 88.90 | 83.05 | 24.14 | 92.91 | 90.16 | 84.19 |
| Marigold [17] | × | 5.99 | 57.90 | 47.13 | 32.63 | 8.46 | 76.33 | 65.90 | 51.52 | 5.72 | 56.79 | 45.87 | 31.26 |
| DA V2 metric + align | × | 5.71 | 62.70 | 48.94 | 32.18 | 12.72 | 77.24 | 68.46 | 54.70 | 5.45 | 62.05 | 48.17 | 31.36 |
| Metric3D + align | × | 3.09 | 43.05 | 29.65 | 16.85 | 8.72 | 76.87 | 64.68 | 47.62 | 2.76 | 41.28 | 27.91 | 15.22 |
| DepthPro + align | × | 4.02 | 53.76 | 40.30 | 23.81 | 6.02 | 64.25 | 55.19 | 40.39 | 3.96 | 53.25 | 39.61 | 23.12 |
| Dust3R [41] | × | 3.70 | 48.57 | 34.16 | 19.53 | 8.69 | 73.40 | 64.03 | 50.18 | 3.34 | 47.21 | 32.56 | 17.93 |
| VGGT [39] | × | 3.70 | 34.05 | 23.44 | 14.58 | 10.78 | 72.22 | 65.12 | 55.83 | 3.32 | 32.27 | 21.34 | 12.24 |
| RAFT-Stereo [27] | × | 4.08 | 17.61 | 14.87 | 12.17 | 9.55 | 67.84 | 59.43 | 47.46 | 3.13 | 13.10 | 10.70 | 8.63 |
| Selective-RAFT [42] | × | 4.05 | 19.48 | 16.64 | 13.57 | 10.08 | 70.02 | 61.79 | 49.64 | 2.92 | 14.94 | 12.38 | 9.93 |
| Selective-IGEV [42] | × | 4.52 | 19.23 | 16.51 | 13.84 | 9.22 | 67.00 | 58.99 | 47.21 | 3.52 | 14.69 | 12.28 | 10.20 |
| MochaStereo [5] | × | 3.79 | 16.77 | 14.24 | 11.77 | 9.18 | 66.64 | 58.10 | 45.78 | 2.82 | 12.25 | 10.11 | 8.30 |
| StereoAnything [13] | × | 4.36 | 24.13 | 19.20 | 14.50 | 10.54 | 73.48 | 63.53 | 49.37 | 3.29 | 20.09 | 15.37 | 11.08 |
| ours | ✓ | 2.43 | 13.84 | 9.98 | 6.91 | 7.32 | 56.77 | 47.83 | 36.45 | 1.76 | 10.06 | 6.54 | 4.08 |

In Table 1, we can observe that monocular depth estimation methods struggle notably with inpainting, replay, and picture illusions, leading to large errors in both disparity and depth spaces. Even when ground-truth alignment is used to convert relative depth to absolute scale, their performance remains significantly worse than stereo methods and ours. As for recent foundation models, such as Dust3R [41] and VGGT [39], they exhibit a strong monocular bias in illusion-rich scenes. In comparison with stereo matching methods depending on explicit correspondence, our method achieves comparable results, indicating that it preserves strong matching constraints. Qualitative results in Figure 5 further illustrate the serious influence of

Table 3: Results of Marigold with/without finetuning on 3D-Visual-Illusion dataset.

| Region | Finetune | EPE↓ | bad2↓ | bad3↓ | AbsRel↓ | $\delta_1$↑ |
|---|---|---|---|---|---|---|
| Illusion | × | 21.16 | 65.67 | 59.67 | 0.45 | 63.65 |
| Illusion | ✓ | 13.67 | 74.82 | 55.20 | 0.28 | 71.04 |
| Non-illusion | × | 7.61 | 49.18 | 39.56 | 0.18 | 79.76 |
| Non-illusion | ✓ | 7.10 | 55.63 | 44.09 | 0.16 | 77.44 |

Table 4: Results of stereo methods on Booster dataset with/without finetuning on 3D-Visual-Illusion data.

| Method | Finetune | Trans | | | NonTrans | | |
|---|---|---|---|---|---|---|---|
| | | EPE↓ | bad2↓ | bad3↓ | EPE↓ | bad2↓ | bad3↓ |
| RAFT-Stereo [27] | × | 9.55 | 67.84 | 59.43 | 3.23 | 13.13 | 10.75 |
| RAFT-Stereo [27] | *Sparse* | 15.36 | 80.34 | 72.34 | 7.12 | 27.47 | 24.01 |
| RAFT-Stereo [27] | *Dense* | 9.24 | 74.10 | 60.67 | 17.39 | 22.48 | 18.96 |
| Selective-IGEV [42] | × | 9.50 | 66.85 | 58.90 | 3.60 | 14.74 | 12.34 |
| Selective-IGEV [42] | *Sparse* | 9.42 | 64.06 | 54.21 | 5.97 | 14.63 | 12.46 |
| Selective-IGEV [42] | *Dense* | 10.39 | 69.65 | 59.40 | 5.32 | 19.00 | 15.64 |

Table 5: Zero-shot generalization on Middlebury dataset at half resolution. Evaluation is conducted in metric disparity space over the entire image, without restricting maximum disparity.

| Metric | Selective-RAFT | Selective-IGEV | MochaStereo | StereoAnything | RAFT-Stereo | Ours |
|---|---|---|---|---|---|---|
| EPE ↓ | 2.34 | 2.59 | 2.66 | 2.89 | 1.92 | **1.50** |
| Bad-2 ↓ | 12.04 | 11.79 | **10.18** | 11.93 | 12.60 | 11.79 |

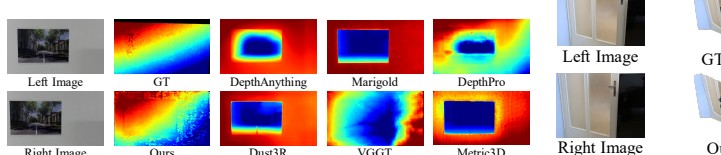

Figure 5: Visualization on our dataset.    Figure 6: Visualization on Booster dataset.

3d visual illusions on SOTA depth estimation methods. More visualization, please refer to our supplemental materials.

The Booster dataset [32] includes many objects with specular and transparent surfaces, making it well-suited for evaluating generalization to mirror illusions. As shown in Table 2, monocular methods achieve the best performance, especially DepthPro. In contrast, binocular methods are easily fooled by mirrors and specular objects. Qualitative results in Figure 6 further show that mirror illusions pose a serious challenge to SOTA binocular methods. See supplemental materials for more visualizations.

Here, we also compare classical binocular and monocular methods with/without finetuning on 3D-Visual-Illusion data. Table 3 shows Marigold's performance on illusion and non-illusion regions of real-world 3D-Visual-Illusion data. We omit DepthAnything V2 results, as its official code fails to converge on the virtual data. This is because its official implementation trains on metric depth, which is sensitive to scale variations, while our virtual data varies significantly in scale. In contrast, the official code of Marigold supports training on affine-invariant depth, ensuring stable learning. Table 3 shows that fine-tuning improves most metrics on illusion regions, suggesting Marigold adjusts its predictions toward planar surfaces, but at the cost of degraded performance on non-illusion regions. This supports our hypothesis that monocular depth models rely on fixed mappings from texture cues (e.g., shape, shadow, perspective, defocus), making it difficult to distinguish illusion textures from real object textures. The worsened bad2 metric in illusion regions indicates incomplete overfitting, while the limited EPE gain in non-illusion regions likely stems from correcting a few extreme outliers rather than consistent improvement.

Table 4 presents the performance of SOTA stereo models with and without finetuning on the 3D-Visual-Illusion dataset. The *Sparse* and *Dense* represent different augmentation strategies used during training. The results show that all stereo models achieve only limited improvement on transparent regions after finetuning, while suffering a significant performance drop on non-transparent regions. This indicates that standard stereo architectures cannot effectively learn from such data. When finetuning on our virtual illusion data, different illusion types are inherently conflicting: mirror illusions rely on spatial context (i.e., monocular priors) for accurate depth estimation, whereas inpainting, picture, replay, and holography illusions deliberately mislead models by distorting these priors. Thus, features learned from mirror illusions are compromised when the model is trained on other illusion types, leading to conflicting learning of monocular priors. Moreover, since we assume a flat plane to rectify disparity during the generation of virtual illusion data, the finetuned stereo models tend to produce overly flat disparities. This results in a slight improvement on transparent glass regions but severe degradation on other non-transparent and non-planar objects.

In addition to illusion scenes, we also present the performance of our model in the Middlebury dataset. We compare our model with several SOTA stereo-based approaches using metric disparity space over the entire image, without restricting the maximum disparity range. Table 5 shows that our model does not degrade in performance on these mundane scenes. On the contrary, it achieves improvements, particularly in terms of EPE.

## 5.2 Ablation Study

We conduct ablation studies on the Booster dataset to evaluate the contribution of each module and various fusion strategies. As shown in Table 6, the baseline stereo-only model performs poorly, indicating that binocular cues alone are insufficient in regions with challenging materials. Introducing

Table 6: Ablation study on the Booster dataset. MF: Monocular Feature, PF: Post Fusion, APF: Adaptive Post Fusion, SF: Stereo Fusion, VLM: Vision-Language Model.

| MF | PF | APF | SF | VLM | EPE ↓ | bad2 ↓ | bad3 ↓ | bad4 ↓ | bad5 ↓ | bad6 ↓ | bad7 ↓ |
|----|----|-----|----|-----|-------|--------|--------|--------|--------|--------|--------|
| | | | | | 15.11 | 80.38 | 72.35 | 66.06 | 61.32 | 57.04 | 52.97 |
| | ✓ | | | | 8.36 | 69.89 | 61.01 | 53.50 | 47.47 | 42.16 | 37.43 |
| ✓ | ✓ | | | | 9.25 | 68.46 | 59.03 | 51.48 | 45.86 | 40.29 | 35.60 |
| ✓ | | ✓ | | | 9.59 | 72.77 | 61.95 | 52.90 | 46.31 | 40.28 | 35.12 |
| ✓ | | | ✓ | | 10.40 | 81.94 | 67.57 | 57.82 | 50.17 | 44.81 | 39.69 |
| ✓ | | | | ✓ | 7.32 | 56.77 | 47.83 | 41.48 | 36.45 | 32.28 | 28.75 |

monocular depth through simple post fusion (PF, where fusion is guided by confidence generated from image features) significantly improves generalization. Incorporating monocular features into the stereo branch (MF + PF) further improves performance on the *bad* metrics, although it slightly degrades the EPE error rate, which indicates better overall geometry but more severe outlier shifts. Adaptive post fusion (APF) employs two independent GRUs to iteratively update the monocular and binocular disparity during fusion, where each other's disparity is used as update guidance. Although this strategy can bring some improvements, it may introduce noise due to inconsistent updates between the two branches. SF uses a single GRU to fuse binocular disparity with fixed monocular disparity, which makes performance worse, highlighting the risks of naïvely reusing uncertain priors. Finally, our full VLM-based fusion approach achieves the best performance, improving the *bad2* metric by over 10 points. This demonstrates the strong reasoning capability of the VLM in handling visually ambiguous regions and its effectiveness in guiding reliable depth fusion. We also evaluate the VLM's confidence by comparing the predicted confidence maps with the disparity error maps, as defined in Equation 14. This analysis is conducted on the Booster dataset under a zero-shot generalization setting. The results show that the error rate of our confidence estimation is approximately $20\%$, demonstrating a strong generalization ability, even in previously unseen illusory scenes.

### 5.3 Discussion

**Illusion Effect on Different Depth Paradigms**: (1) Monocular estimation relies on texture-based cues (e.g., shape, perspective, shadow, defocus) learned from RGB image. When these cues are artificially simulated on flat surfaces (inpainting, pictures, replays, holograms), the model is easily misled, producing incorrect depths. Mirror illusions, however, can often be resolved through scene-level context learned from large-scale data. (2) Stereo estimation instead depends on pixel-wise correspondence. In mirror scenes, reflections overlap with real surfaces, causing ambiguous matches and depth errors. For inpainting, pictures, replays, and holography, stereo matching remains effective due to cross-view texture consistency.

**Limitations and Future Work**: (1) The virtual data generation pipeline relies on manual semantic segmentation, which is labor-intensive and time-consuming. Given the lack of 3D visual illusion data, and the fact that existing detectors, segmentation models, and VLMs are often fooled, even humans will be fooled in complex cases, manual collection remains a practical step at this stage. Developing an automatic pipeline is an important future direction. (2) The real-world subset currently covers only a limited range of illusions (inpainting, picture, and replay). We plan to extend it to broader types and more diverse real-world scenes. (3) The VLM-driven fusion is effective but computationally expensive. Designing lighter, more efficient fusion methods is worth further exploration. (4) Our study focuses on pure illusions without compositing. Future challenges include combinations of multiple illusions, semantically ambiguous objects, and entirely novel illusion types.

## 6 Conclusion

In this paper, we introduce the 3D-Visual-Illusion dataset, a large-scale benchmark for evaluating the depth estimation models under 3D visual illusions. The dataset covers diverse illusion types and scene categories, including indoor and outdoor settings, as well as both virtual and real-world data. Our experiments show that state-of-the-art models are easily fooled by various illusions, each exhibiting distinct failure modes. Monocular methods act as generative models, mapping texture cues to 3D geometry, and thus can be deceived by carefully simulated textures. In contrast, stereo methods serve as discriminative models that rely on pixel-wise correspondence, which breaks down when multiple objects project onto the same pixels. Finally, we introduce a VLM-driven monocular–stereo fusion model, which leverages commonsense reasoning from a vision-language model to assess cue reliability and achieve more robust depth estimation.

**Acknowledgment** This work was supported by the Shenzhen Science and Technology Program under Grant No. JCYJ20241202130548062, the Natural Science Foundation of Shenzhen under Grant No. JCYJ20230807142703006, the Natural Science Foundation of China (NSFC) under Grants No. 62172041 and No. 62176021.

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

# Appendix

## A    3D Visual Illusion Dataset

### A.1    Video Collection

We collect a large amount of videos from web-source data and generative models, covering five types of illusions: inpainting illusion (e.g., inpainting on a wall/floor), picture illusion (e.g., picture printed/drawn on a paper), replay illusion (e.g., video replayed on different screens), holography illusion, and mirror illusion (e.g., specular or transparent surfaces), as shown in Figure 7. When collecting videos from generative models, we observe four important considerations in the design of text prompts. (1) Level of Detail in Prompts: Overly fine-grained control in prompts often leads to physically unrealistic results, such as requiring the object in the mirror to maintain the same pose as its real-world counterpart, enforcing perfect mirror symmetry, or specifying excessive positional details. Instead, less detailed scene descriptions tend to produce more physically accurate and realistic results. (2) Challenges with Dynamic Objects: Generating videos with dynamic objects proves particularly difficult. The virtual image in the mirror and the real-world objects often exhibit motion inconsistencies. As a result, we focus primarily on static scenes or those with only slight object movement. (3) Layout Complexity: Complex scene layouts frequently lead to mismatches between the mirror world and the real world, causing spatial inconsistencies. (4) Camera Motion: To ensure a stable and realistic scene, the camera is required to pan slowly. Excessive camera movement may result in abrupt rotations or scene transitions, disrupting the illusion.

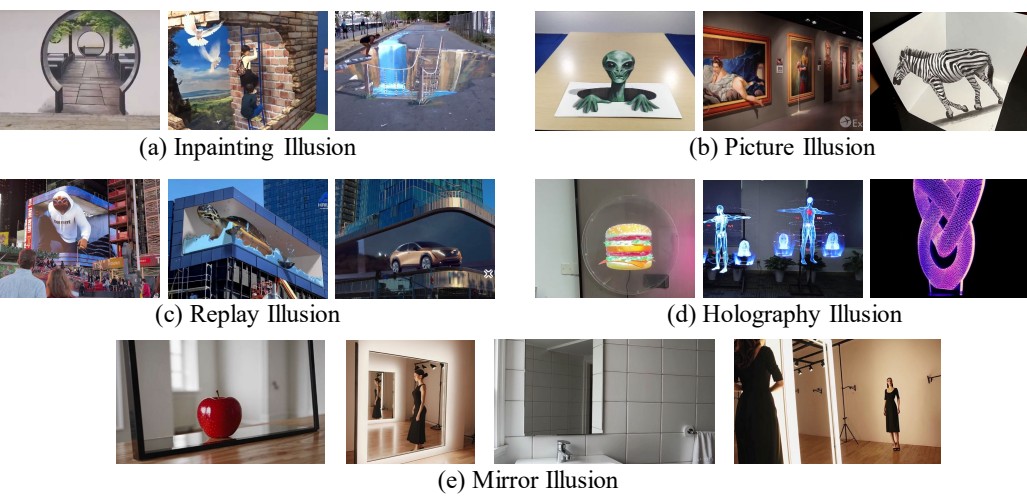

(a) Inpainting Illusion

(b) Picture Illusion

(c) Replay Illusion

(d) Holography Illusion

(e) Mirror Illusion

Figure 7: The visualization of 3D visual illusions.

### A.2    Depth Geneation

To mitigate the impact of noise in plane fitting, we adopt RANSAC for robust plane estimation:

$$\min_{\alpha,\beta,\delta,\gamma} \quad \sum_{i=1}^{N}(\alpha \cdot u_i + \beta \cdot v_i + \delta \cdot d_i + \gamma)^2, \tag{11}$$
$$subject \ to \ \alpha^2 + \beta^2 + \delta^2 = 1.$$

$(\alpha, \beta, \delta, \gamma)$ are the plane parameters, $(u, v)$ is image plane coordinate and $d$ is disparity. As illustrated in Algorithm 1,, we randomly sample three points to define a candidate plane at each iteration of RANSAC. The plane normal $(\alpha, \beta, \delta)$ is computed as the cross product of vectors formed by these three points, and the offset $\gamma$ is derived by substituting one point into $\alpha \cdot u + \beta \cdot v + \delta \cdot d + \gamma = 0$. We then compute the distance from each point in the support region to the candidate plane to determine the inliers. After all iterations, the candidate plane with the largest number of inliers is selected, which are taken as the best inlier set. The plane parameters $(\alpha, \beta, \delta, \gamma)$ are then estimated by computing the

eigenvector corresponding to the smallest eigenvalue of the covariance matrix constructed from the best inlier set. We also present visualizations of the rendered and rectified depth images in Figure 8 and 9. After applying plane fitting for rectification, the resulting depth map becomes smoother and more geometrically accurate.

---

**Algorithm 1** RANSAC Plane Fitting

---

**Require:** Point set $\mathbf{P} = \{(u_i, v_i, d_i)\}_{i=1}^N \in \mathbb{R}^{N \times 3}$, inlier threshold $\tau_d$, sub-sample size per iteration $M$, max iterations $T_p$
**Ensure:** Optimal plane parameters $\pi^* = [\alpha, \beta, \delta, \gamma]$
 1: Initialize: $best\_score = 0$, $best\_plane = \mathbf{0}$, $best\_inliers = \emptyset$
 2: **for** $t = 1$ **to** $T_p$ **do**
 3:     Randomly sample $M$ sets of 3-point tuples:
        $Q = \{(u_i^0, v_i^0, d_i^0), (u_i^1, v_i^1, d_i^1), (u_i^2, v_i^2, d_i^2)\}_{i=1}^M \in \mathbb{R}^{M \times 3 \times 3}$
 4:     **for** $b = 1$ **to** $M$ **do**
 5:       $\mathbf{v}_{10} = (u_i^1, v_i^1, d_i^1) - (u_i^0, v_i^0, d_i^0)$, $\mathbf{v}_{20} = (u_i^2, v_i^2, d_i^2) - (u_i^0, v_i^0, d_i^0)$
 6:       $\mathbf{n}_b = \mathbf{v}_{10} \times \mathbf{v}_{20}$                                            {Normal via cross product}
 7:       $d_b = -\mathbf{n}_b^\top [u_i^1, v_i^1, d_i^1]$
 8:       $\pi_t[b] = [\mathbf{n}_b^\top, d_b]$
 9:     **end for**
10:     $\mathbf{D}_t = \pi_t[\mathbf{P}, \mathbf{1}]^\top / \|\pi_t[:, 0:3]\|_2$                                    {Batch distance computation}
11:     $\mathbf{M}_t = \|\mathbf{D}_t\| < \tau_d$
12:     $\mathbf{c}_t = \text{sum}(\mathbf{M}_t, \text{dim=1})$
13:     $k = \arg\max \mathbf{c}_t$, $c_{\max} = \mathbf{c}_t[k]$
14:     **if** $c_{\max} > best\_score$ **then**
15:       $best\_score = c_{\max}$
16:       $best\_plane = \pi_t[k]$
17:       $best\_inliers = \mathbf{M}_t[k]$
18:     **end if**
19: **end for**
20: **Refinement via Eigen Decomposition**
21: $\mathbf{P}_{\text{inliers}} = \mathbf{P}[best\_inliers]$
22: $\mathbf{S} = [\mathbf{P}_{\text{inliers}}, \mathbf{1}]^\top [\mathbf{P}_{\text{inliers}}, \mathbf{1}]$
23: $(\mathbf{W}, \mathbf{V}) = \text{eigh}(\mathbf{S})$
24: $\pi^* = \mathbf{V}[:, 0]$
25: **return** $\pi^*$

---

### A.3 Right Image Geneation

The right-view images for generative-model videos are directly rendered using Gaussian Splatting (GS). For web-sourced videos, right views are generated by warping the left images using monocular disparity. As shown in Algorithm 2, we generate a right-view image $\hat{I}_R$ from a given left-view image $I_L$ and disparity map $D$. It begins by estimating an appropriate disparity scaling factor $s$ via binary search, ensuring that a sufficient proportion of the projected pixels fall within valid image bounds. Using the computed $s$, pixel coordinates are mapped from the left to the right view, with invalid coordinates filtered out. An initial right-view image is synthesized by transferring valid pixel values based on the mapping. Finally, image inpainting is applied to fill missing regions, resulting in the completed right-view image $\hat{I}_R$. The algorithm outputs both $\hat{I}_R$ and the estimated scaling factor $s$. We also present the visualization of the initial warped image and the inpainted image in Figure reffig: vis web source. The inpainting process effectively fills in the missing regions, resulting in a more complete and visually coherent right-view image.

### A.4 Real-world Data

#### A.4.1 Camera System

We collect real-world data using a stereo camera (ZED Mini) and a LiDAR-based depth sensor (Realsense L515). The sensors are rigidly mounted and calibrated using a checkerboard to ensure

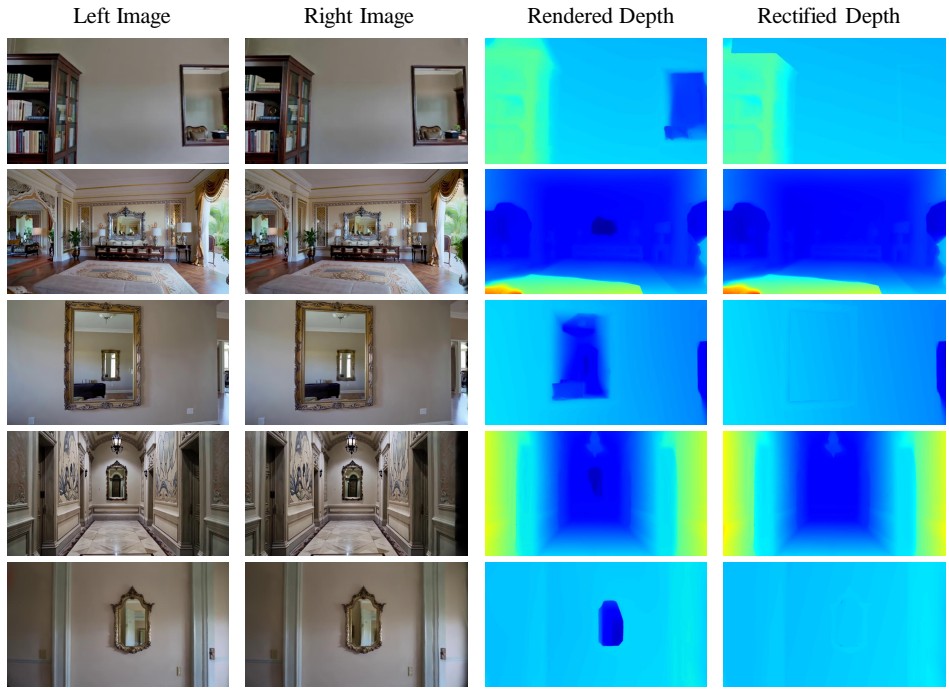

Figure 8: The visualization of results for video from generative models.

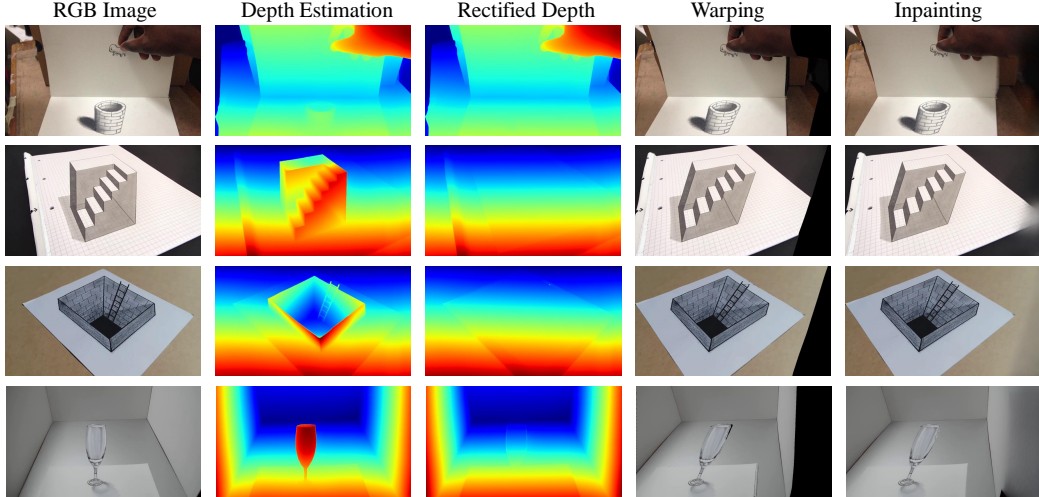

Figure 9: The visualization of results for web-source video.

accurate alignment, as shown in Figure 10. The ZED Mini captures RGB images, while the L515 provides depth maps. The intrinsic and extrinsic parameters of both cameras are obtained through calibration. The calibration process involves capturing multiple images of the checkerboard pattern from different angles and distances, allowing for accurate estimation of the camera parameters.

### A.4.2    Depth Map Projection

The L515 depth map is warped to the ZED left camera to construct the ground-truth depth. As shown in Algorithm 3, the process begins by upsampling the depth map and scaling the intrinsic matrix accordingly. 3D points are then computed and transformed from the L515 frame to the ZED frame using calibrated extrinsics, followed by projection onto the ZED image plane. The resulting depth values are splatted to the ZED image grid, and missing regions are filled using inpainting and

---

**Algorithm 2** Right Image Generation

---

**Require:** Left image $I_L \in \mathbb{R}^{H \times W \times 3}$, disparity map $D \in \mathbb{R}^{H \times W}$, valid pixel threshold $\theta = 0.9$, maximum iterations $T_g$

**Ensure:** Synthesized right image $\hat{I}_R$, scaling factor $s$

1: **Step 1: Compute Scaling Factor**
2: Initialize: $l = 0, r = W/(4 \cdot \max(D)), \epsilon = 10^{-6}, t = 0$
3: **while** $|r - l| > \epsilon$ **and** t $< T_g$ **do**
4:     $t = t + 1$
5:     $s = (l + r)/2$
6:     Coordinate projection: $U' = U - s \cdot D$
7:     Compute valid pixel ratio: $\eta = \frac{1}{HW} \sum \mathbb{I}(U' \in [0, W))$
8:     **if** $\eta \geq \theta$ **then**
9:       $l = s$
10:     **else**
11:       $r = s$
12:     **end if**
13: **end while**
14: Final scaling factor: $s = (l + r)/2$
15: **Step 2: Image Coordinate Mapping**
16: Generate coordinate grid: $(u, v) = \text{MESHGRID}(0 : W - 1, 0 : H - 1)$
17: Compute projected coordinates: $u' = u - s \cdot D(u, v)$
18: Quantize coordinates: $\hat{u}' = \{\lfloor u' \rfloor, \lceil u' \rceil\}$
19: Filter invalid coordinates: $\{\hat{u}' \mid \hat{u}' \geq 0 \text{ and } \hat{u}' < W\}$
20: **Step 3: Right View Image Synthesis**
21: Initialize: $I_R = \mathbf{0}^{H \times W \times 3}$
22: **for** each pixel $(u, v)$ **do**
23:     Generate initial right-view image $I_R$: $I_R(u', v) = I_L(u^*, v), u^* = \arg\max_u \{d_{(u,v)} \mid u - s \cdot D_{(u,v)} = u'\}$.
24: **end for**
25: **Step 4: Image Completion** Perform inpainting on $I_R$ to fill invalid regions and obtain the final right-view image $\hat{I}_R$
26: **return** $\hat{I}_R, s$

---

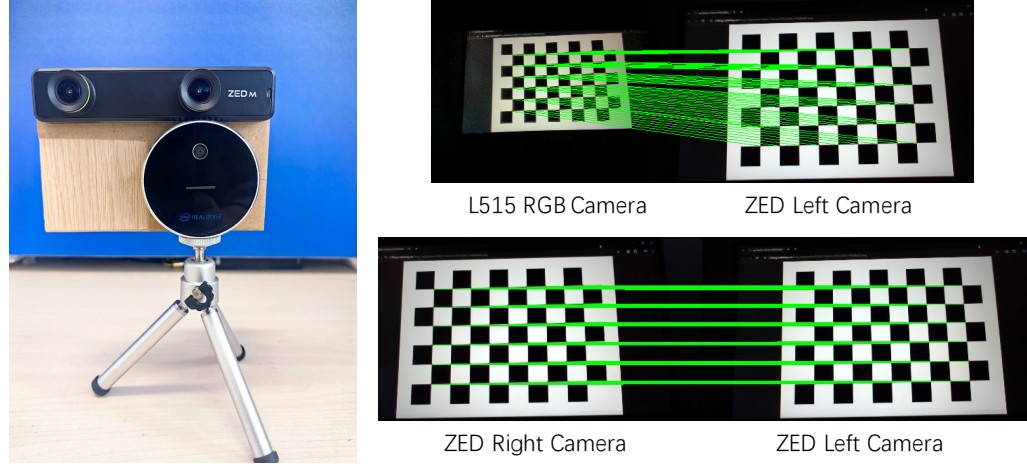

Figure 10: Camera System and Calibration Visualization

guided filtering. To ensure consistency, a backward reprojection step verifies each pixel's validity by comparing it with the original L515 depth. Finally, noise is suppressed using median filtering, and valid depth values are converted into disparities based on the ZED stereo baseline and focal length.

---

**Algorithm 3** Depth Map Reprojection

---

**Require:** Depth map from L515 camera $\mathcal{Z}_L \in \mathbb{R}^{H_L \times W_L}$, RGB image from ZED left camera $I_Z \in \mathbb{R}^{H' \times W' \times 3}$, intrinsic matrix of L515 $K_L \in \mathbb{R}^{3 \times 3}$, rotation matrix $R \in \mathbb{R}^{3 \times 3}$ and translation matrix $T \in \mathbb{R}^{3 \times 1}$ from L515 to ZED-left, upsampling factor $s = 3$, ZED stereo baseline $B$, and ZED focal length $F$

**Ensure:** Disparity map of ZED left camera $D \in \mathbb{R}^{H' \times W'}$
 1: **Step 1: Depth Upsampling**
 2: $\tilde{\mathcal{Z}}_{\mathrm{L}} = \mathrm{resize}(\mathcal{Z}_L, \mathrm{scale} = s, \mathrm{interp} = \texttt{NEAREST})$
 3: $\tilde{K}_{\mathrm{L}} = s \cdot K_L$
 4: **Step 2: Coordinate Transformation**
 5: **for** each pixel $(u_L, v_L)$ in $\tilde{\mathcal{Z}}_{\mathrm{L}}$ **do**
 6: $\quad [x_Z, y_Z, z_Z] = R \cdot \tilde{\mathcal{Z}}_{\mathrm{L}} \cdot \tilde{K}_{\mathrm{L}}^{-1} \cdot [u_L, v_L, 1]^T + T$
 7: $\quad [u_Z, v_Z, 1] = K_Z \cdot [x_Z/z_Z, y_Z/z_Z, 1]$
 8: **end for**
 9: **Step 3: Depth Projection**
10: Initialize $\mathcal{Z}_{\mathrm{Z}} = \infty^{H' \times W'}$
11: **for** each projected point $(u_Z^i, v_Z^i, z_Z^i)$ **do**
12: $\quad (u_1, v_1) = (\lfloor u_Z^i \rfloor, \lfloor v_Z^i \rfloor), (u_2, v_2) = (\lceil u_Z^i \rceil, \lceil v_Z^i \rceil)$
13: $\quad$ **for** $(u, v) \in \{(u_1, v_1), (u_1, v_2), (u_2, v_1), (u_2, v_2)\}$ **do**
14: $\quad\quad$ **if** $(u, v)$ is within image bounds **then**
15: $\quad\quad\quad \mathcal{Z}_{\mathrm{Z}}(v, u) = \min(\mathcal{Z}_{\mathrm{Z}}(v, u), z_Z^i)$
16: $\quad\quad$ **end if**
17: $\quad$ **end for**
18: **end for**
19: **Step 4: Hole Filling**
20: $\mathcal{M}_{\mathrm{invalid}} = (\mathcal{Z}_{\mathrm{Z}} == \infty), \mathcal{Z}_{\mathrm{Z}} = \mathcal{Z}_{\mathrm{Z}} \odot \neg\mathcal{M}_{\mathrm{invalid}} + \mathbf{0} \odot \mathcal{M}_{\mathrm{invalid}}$
21: $\mathcal{M}_{\mathrm{small}} = \mathrm{connectedComponents}(\mathcal{M}_{\mathrm{invalid}}, \mathrm{area\_th} = 100)$
22: $\mathcal{Z}_{\mathrm{repair\_small}} = inpaint(\mathcal{Z}_{\mathrm{Z}}, \mathcal{M}_{\mathrm{small}}), \mathcal{Z}_{\mathrm{repair\_all}} = inpaint(\mathcal{Z}_{\mathrm{Z}}, \mathcal{M}_{\mathrm{invalid}})$
23: $\mathcal{Z}_{\mathrm{repair\_all}} = \mathrm{guidedFilter}(I_Z, \mathcal{Z}_{\mathrm{repair\_all}}, \mathrm{radius} = 5, \epsilon = 1e-3)$
24: $\mathcal{Z}_{\mathrm{repair}} = \mathcal{Z}_{\mathrm{Z}} \odot \neg\mathcal{M}_{\mathrm{invalid}} + \mathcal{Z}_{\mathrm{repair\_all}} \odot \neg(\mathcal{Z}_{\mathrm{repair\_small}} == 0) \odot \mathcal{M}_{\mathrm{invalid}}$
25: **Step 5: Backward Reprojection for Invalid Region Detection**
26: **for** each pixel $(u_Z, v_Z)$ in $\mathcal{Z}_{\mathrm{repair}}$ **do**
27: $\quad [x_{Z \to L}, y_{Z \to L}, z_{Z \to L}] = R^{-1} \cdot (z_Z \cdot K_Z^{-1} \cdot [u_Z, v_Z, 1] - T)$
28: $\quad [u_{Z \to L}, v_{Z \to L}, 1] = \frac{1}{z_{Z \to L}} \cdot K_L[x_{Z \to L}, y_{Z \to L}, z_{Z \to L}]$
29: $\quad$ **if** $\mathcal{Z}_L(v_{Z \to L}, u_{Z \to L}) == 0$ or $\|\mathcal{Z}_L(v_{Z \to L}, u_{Z \to L}) - z_{Z \to L}\| > \tau$ **then**
30: $\quad\quad \mathcal{Z}_{\mathrm{repair}}(u_Z, v_Z) = 0$
31: $\quad$ **end if**
32: **end for**
33: **Step 6: Noise Suppression**
34: $\mathcal{Z}_{\mathrm{smooth}} = \mathrm{medianFilter}(\mathcal{Z}_{\mathrm{repair}}, \mathrm{size} = 3)$
35: $\mathcal{M}_{\mathrm{noise}} = |\mathcal{Z}_{\mathrm{repair}} - \mathcal{Z}_{\mathrm{smooth}}| > 0.03$
36: $\mathcal{Z}_{\mathrm{final}} = \mathcal{Z}_{\mathrm{repair}} \odot \neg\mathcal{M}_{\mathrm{noise}} + \mathbf{0} \odot \mathcal{M}_{\mathrm{noise}}$
37: **Step 7: Disparity Computation**
38: $D = B \cdot F / \mathcal{Z}_{\mathrm{final}}$
39: **return** $D$

---

## B  VLM-Driven Monocular-Stereo Fusion Model

Inspired by the strong reasoning ability of vision-language models (VLMs), as shown in Figure 11. We propose a VLM-driven monocular-stereo fusion model that leverages commonsense knowledge to assess the reliability of monocular and stereo depth in different regions, enabling effective depth fusion.

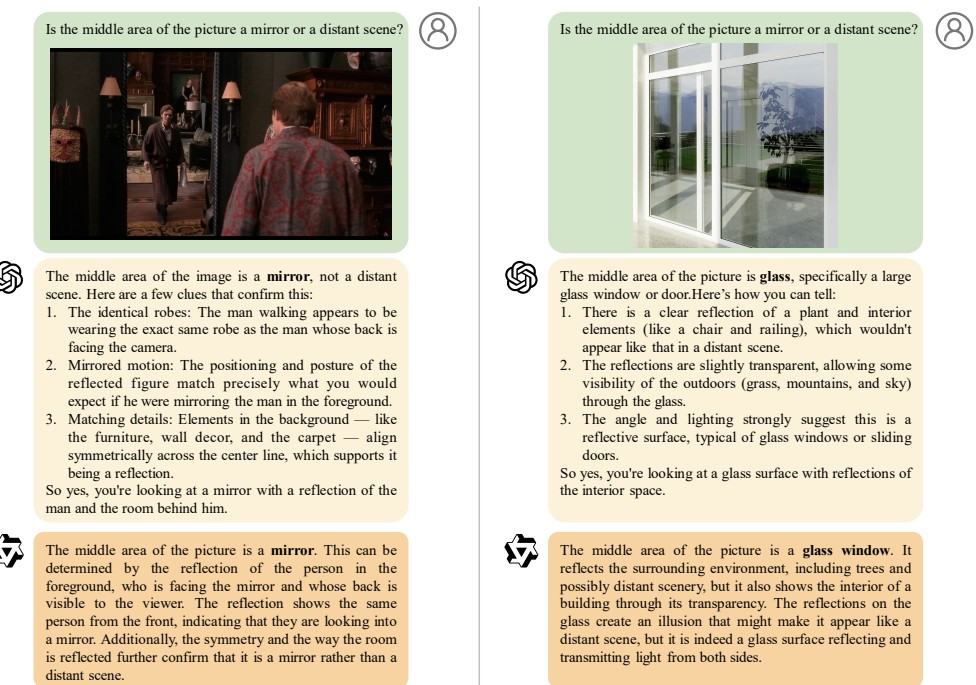

Figure 11: Visualization of commonsense knowledge from VLMs. The left image shows a scene with a mirror. The right image depicts a scene with a transparent object, where the background is visible through the object.

### B.1  Loss

The supervised loss function consists of two main components: one ($\mathcal{L}_d$) for the disparity maps and the other ($\mathcal{L}_c$) for the confidence map:

$$\mathcal{L} = \mathcal{L}_d + w\mathcal{L}_c, \tag{12}$$

where $w$ is a manually set weighting factor for balancing the confidence map loss.

For disparity supervision, we use the $L_1$ loss to supervise each iteratively updated disparity $D_s^t$, the aligned monocular disparity $\tilde{D}_m$, and the final predicted disparity $D_f$. The loss function is defined as:

$$\mathcal{L}_d = \sum_{t=1}^{T} \gamma_d^{T+2-t} ||D_s^t - D_G||_1 \\ + \gamma_d||\tilde{D}_m - D_G||_1 + ||D_f - D_G||_1. \tag{13}$$

Here, $D_G$ denotes the ground-truth disparity, and $\gamma_d$ is a weighting coefficient to balance contributions from intermediate predictions.

For confidence map supervision, we adopt the Focal Loss, where the ground-truth for confidence is derived based on the disparity difference between the final stereo prediction $D_s^T$ and the ground-truth

$D_G$:

$$\mathcal{L}_c = \frac{1}{N} \sum_i \alpha_c \cdot \left(1 - e^{-\mathcal{L}_b(i)}\right)^{\gamma_c} \cdot \mathcal{L}_b(i),$$

$$\mathcal{L}_b = -\bar{I}_c \log I_c - (1 - \bar{I}_c) \log(1 - I_c), \tag{14}$$

$$\bar{I}_c = \mathbb{I}(\text{Interpolate}(|D_G - D_s^T|, \text{scale} = \frac{1}{4}) < \frac{5}{4}),$$

In this formulation, $\alpha_c$ and $\gamma_c$ are the hyperparameters of the Focal Loss, $\mathbb{I}$ is the indicator function, and Interpolate denotes the downsampling operation that resizes the supervision signal to $\frac{1}{4}$ of the original resolution, matching the resolution used in intermediate network outputs.

## C  Experiments

### C.1  Evaluation Metric

We evaluate model performance in both disparity space and depth space. In disparity space, two commonly used metrics are adopted. (1) End-Point Error (EPE): $EPE = \frac{1}{N} \sum_i |r_i - \bar{r}_i|$, where $r$ and $\bar{r}$ denote the predicted and ground-truth disparity values, respectively. EPE measures the average absolute disparity error in pixels. (2) Bad-$x$ Error: bad-$x = \frac{1}{N} \sum_i \mathbb{I}(|r_i - \bar{r}_i| > x)$, which indicates the percentage of pixels where the disparity error exceeds $x$ pixels. This metric is especially useful for evaluating the robustness of the model under boundary conditions. In depth space, four standard evaluation metrics are employed. (1) Absolute Relative Error (AbsRel): $AbsRel = \frac{1}{N} \sum_i \frac{|r_i - \bar{r}_i|}{\bar{r}_i}$, which evaluates the relative difference between predictions and ground-truth values, normalized to mitigate the impact of scale and unit differences, making it suitable for datasets with diverse depth ranges. (2) Root Mean Squared Error (RMS): $RMS = \sqrt{\frac{1}{N} \sum_i (r_i - \bar{r}_i)^2}$. (3) Log10 Error: $log10 = \frac{1}{N} \sum_i |\log_{10}(r_i) - \log_{10}(\bar{r}_i)|$. (4) Threshold Accuracy ($\delta_1$): $\delta_1 = \frac{1}{N} \sum_i \mathbb{I}\left(\max\left(\frac{y_i}{\bar{y}_i}, \frac{\bar{y}_i}{y_i}\right) < 1.25\right)$, which measures the proportion of pixels for which the predicted depth falls within a certain ratio (e.g., 1.25) of the ground truth. The model is initially trained on the SceneFlow dataset, then fine-tuned on the 3D-Visual-Illusion training set, and finally evaluated on the 3D-Illusion test set and the Booster training set.

### C.2  Implementation Details

For dataset construction, we use Qwen2-VL-72B [1, 40] to perform initial screening, reducing the dataset from 5226 videos with 52M frames to 4,519 videos with 1.4M frames. We then built a Flask[37]-based web tool to manually reduce data to 1384 videos with 236k frames. We further developed a more convenient flask-based web app for SAM2 [34] to acquire the semantic mask of illusion and support regions. During the semantic segmentation, we delete frames with redundant content and illusions imperceptible to humans, reducing data from 236k frames to 176,530 frames. Later, we rectify the depth values of illusion regions from the reference of support regions, which is further used to generate the right images for web-source data. Besides web-source data, We also use large generative models to generate 234 videos with 2382 frames. The video generation is achieved via Sora [30] and Kling [21], and a small part of the data is generated from HunyuanVideo [20]. We then use InstantSplat [6], DUSt3R [41], and GS [18] to generate the right images and depth map, followed by similar depth post-processing.

As for our VLM-driven monocular-stereo fusion network, we benefit from the vision and language foundation model and use Depthanything V2 [45, 46] as a pre-trained monocular model, QwenVl2-7B [40, 1] as pre-trained VLM, and FLUX [22] as a diffusion model. We use Lora [15] to fine-tune the last layer of QwenVl2-7B and the Q$$V projection layer of FLUX on $4\times$ H100 with a batch size of 6 on each GPU. The entire training takes almost 20 days.

### C.3  Prompts

In dataset construction, we design a prompt to prefilter bad frames using Qwen2-VL-72B [1, 40] due to the large amount of videos collected from the Internet. The prompts are as follows:

Reply to me in the format of a string concatenating 'yes' or 'no' with ','. Each 'yes or 'no' is an answer to each following question. Does this image feature any flat artistic creation of landscapes where the surface of the creation is flat and has no ups and downs? Does this image contain any areas with perspective illusions? Does this image contain any optical-illusion graffiti or artwork? Does this image contain any transparent or high-reflective areas? Does this image show a display screen playing 3D objects or scenes? Does the image contain areas that make you mistake them for 3D objects? Does this image contain excessive watermarks or captions that seriously affect its quality? Does this image contain small watermarks or captions in the corners? Is this image too blurry? Are most regions of the artistic creation covered by a single/two hands? Is this image a software interface? Is only the figure of the artist clear, but the others are blurry, like artwork, screen, or areas that make you mistake them for 3D objects?

We use the answer from Qwen2-VL-72B to filter out bad frames. We reduce the data from 5,226 videos and 52 million frames to 4,519 videos and 1.4 million frames.

In addition to web-sourced data, we also use videos produced by generative models, resulting in 234 videos comprising a total of 2,382 frames. The primary generative models used are Sora and Kling, with a small portion of the data sourced from HunyuanVideo [20]. The initial prompts were generated using ChatGPT, with the prompt used for generation as follows:

Please provide 100 unique and detailed bilingual (Chinese and English) prompts, each with an index number, for generating text-to-video scenes that include mirror reflections. The prompts must meet the following requirements: 1. Specify the mirror type and describe the entities in the scene, the overall layout, and their spatial relationship to the mirror. 2. Include a diverse range of mirror types: dressing mirrors, vanity mirrors, full-length mirrors, bathroom mirrors, car rearview mirrors, polished stainless steel, etc. 3. Ensure varied scene distributions: residential settings, commercial spaces, and public areas. 4. The combination of mirror type and scene context must be reasonable (e.g., polished stainless steel is appropriate in a kitchen but not in a study). 5. Entity configuration: some scenes should include people in front of the mirror (e.g., a woman combing her hair or a customer trying on clothes), some should feature objects (e.g., plants, cosmetics, books), and others should show only the mirror reflecting surfaces like walls. 6. Each prompt must describe the physical correspondence between the real object and its reflection. 7. Avoid overly complex layouts in individual scenes. 8. Ensure a balance of richly textured and minimally textured elements within the same scene. 9. All objects in the scene must remain static, with only slow camera panning; descriptions implying motion (e.g., "a moving car") are inappropriate. 10. Descriptions should be as precise and detailed as possible.

The generated prompts were subsequently refined to avoid producing low-quality video outputs, as pointed in Section A.1. Below are some examples of the prompts:

Generate a video showing a cozy, modern living room. A single minimalist-designed mirror is mounted on the wall, with clearly defined edges and realistic reflections. The scene combines intricate furniture textures with a monochromatic background, and the camera pans slowly.

Generate a video set in a creative art space. A uniquely shaped mirror hangs on the wall, featuring accurate reflections and distinct boundaries. The scene includes complex graffiti textures and smooth surfaces, with slow camera panning.

A static and art-deco inspired living room with a framed mirror above a tufted velvet sofa, reflecting physical laws accurately, geometric patterns, sleek metal finishes, and glamorous lighting. Realistic, glamorous lighting, retro.

> A static and rustic farmhouse dining area with a reclaimed wood-framed mirror on a weathered brick wall, highlighting a crisp realistic reflection, a sturdy wooden table, vintage chairs, and warm pendant lighting. Realistic, warm lighting, rustic.

Our VLM-driven monocular-stereo fusion framework employs Depthanything V2 [45, 46] as the pre-trained monocular network, QwenVl2-7B [40, 1] as the pre-trained visual-language network, and FLUX [26, 31] as the flow matching network. The language prompt for the pre-trained visual-language network is:

> Are there any transparent or reflective objects? Like mirror, glass, window, showcase, and so on? If true, reply to me with the list of corner coordinates of each object in the format of (x1,y1,x2,y2,x3,y3,x4,y4) in the image. If false, reply with an empty list of corners.

The language prompt for the pre-trained flow matching network is:

> Using the provided features extracted by QwenVL2, generate a binary segmentation mask for the image. Highlight all transparent or reflective objects (e.g., mirrors, glass, windows, showcases) in white (255), while marking all other regions in black (0).

## C.4 Computational Cost

The detailed inference metrics, including runtime and memory consumption, are presented in Table 7. All experiments were conducted on a single NVIDIA H100 GPU with an input resolution of $1920 \times 1080$. The majority of the computational cost arises from the VLM part.

| Model | Memory Usage | Inference Time (per iteration) |
|---|---|---|
| RAFT-Stereo | 5610 MB | 0.87 s/it |
| DepthAnything V2 | 3584 MB | 0.18 s/it |
| Ours | 53959 MB | 4.77 s/it |

Table 7: Comparison of memory usage and inference time across models.

## C.5 Visualization

We present more visualization on the 3D-Visual-Illusion dataset and Booster dataset in Figure 12, 13, and 14. The results demonstrate that our method can effectively handle various types of visual illusions. The depth maps generated by our model exhibit high fidelity and accuracy, even in challenging scenarios with complex visual illusions. The depth maps from VGGT and Dust3R mirror the significance of fusing monocular priors and multi-view matching.

We also present the visualization of 3D detection on the real data of the 3D-Visual-Illusion dataset in Figure 15. We obtain the results from YOLO3D [29], and the results show that 3D visual illusions can seriously affect the performance of 3D detection. We believe that the 3D visual illusion will become more and more important as the vision foundation models become more and more powerful, especially when used in downstream applications, like 3D detection, occupancy and planning.

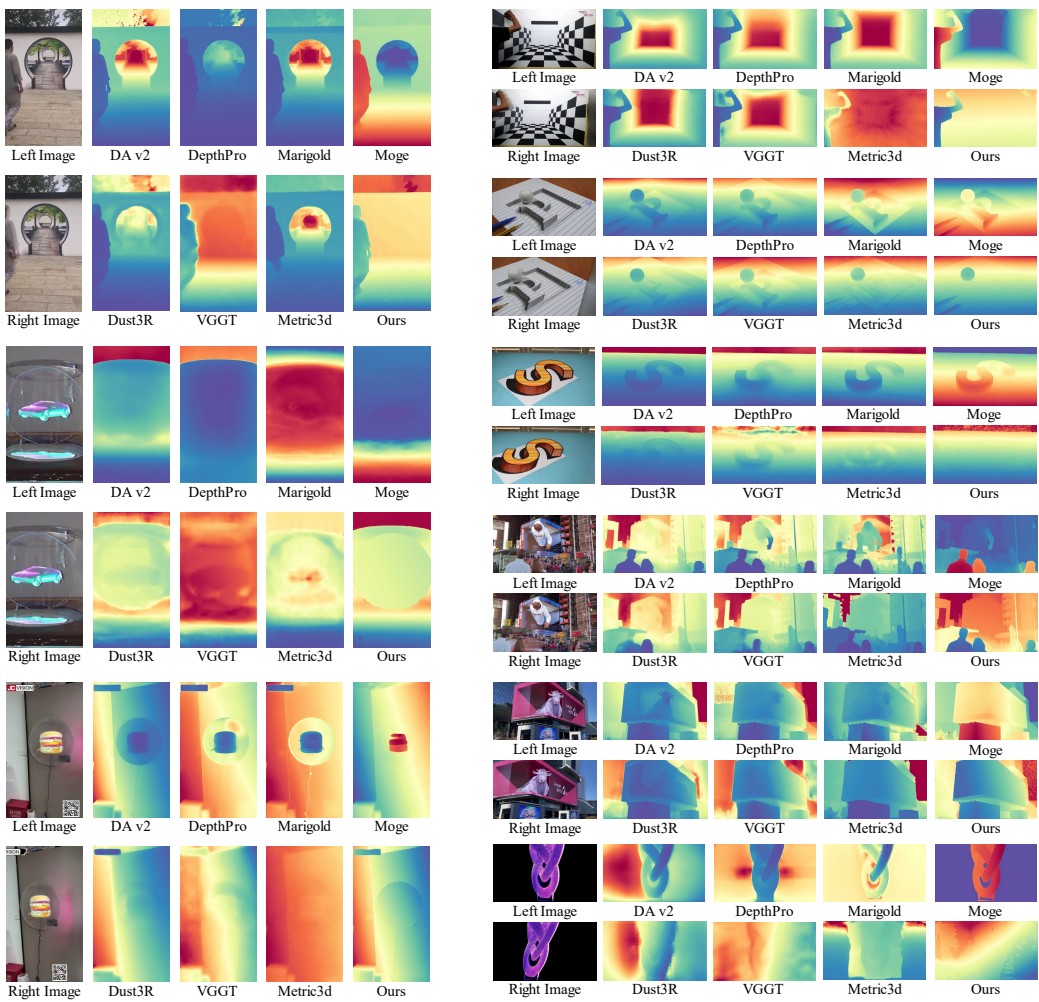

Figure 12: The visualization of results on virtual data of the 3D-Visual-Illusion dataset.

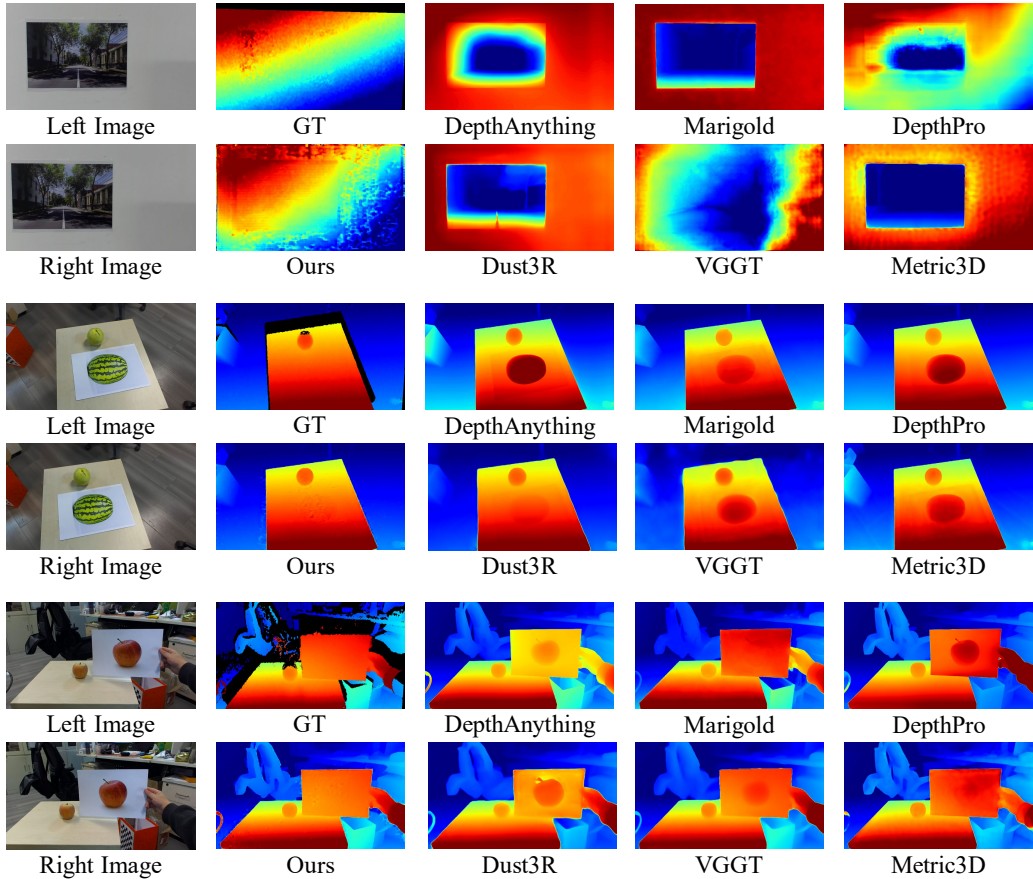

Figure 13: The visualization of results on real data of the 3D-Visual-Illusion dataset.

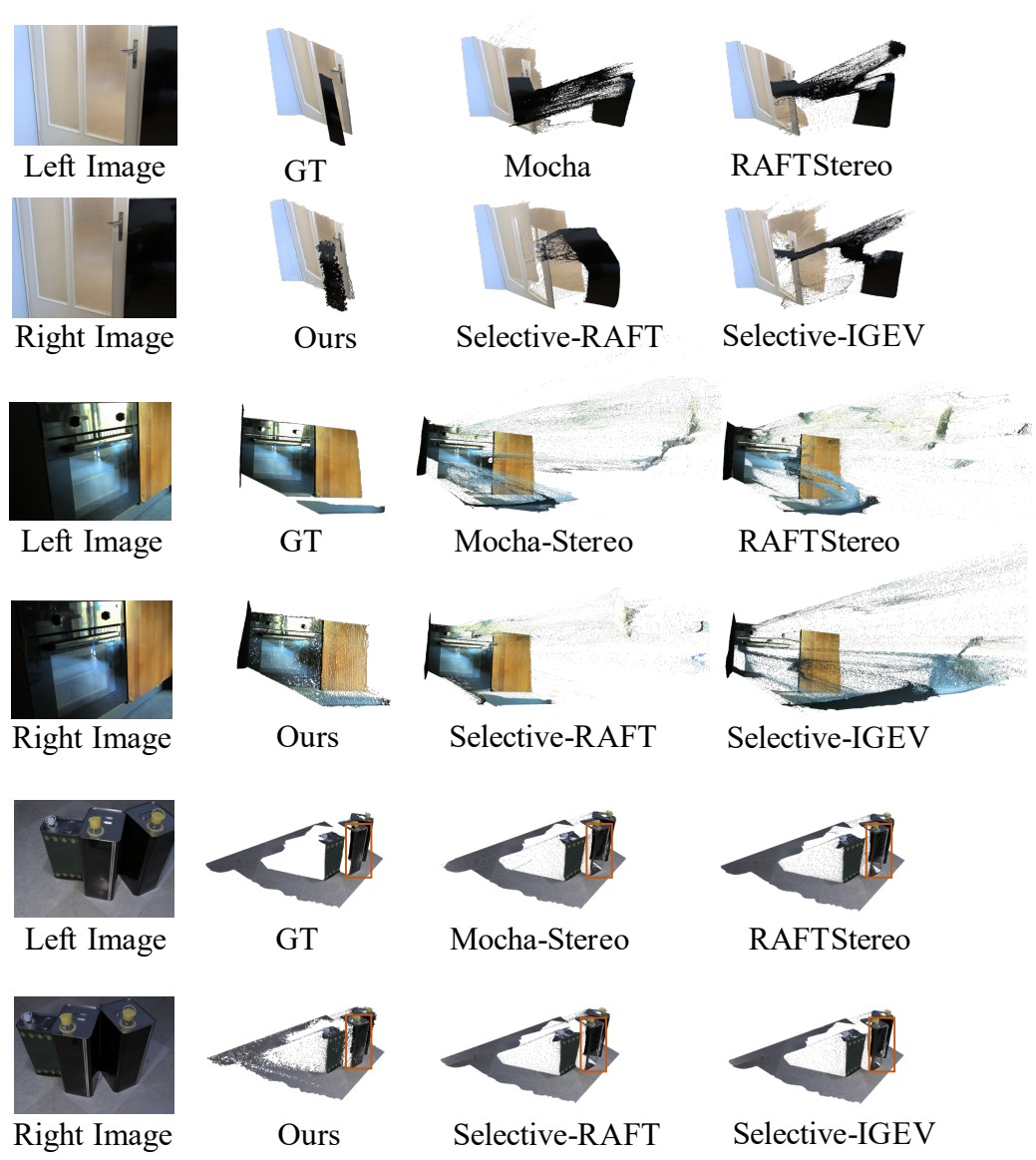

Figure 14: The visualization of results on the Booster dataset.

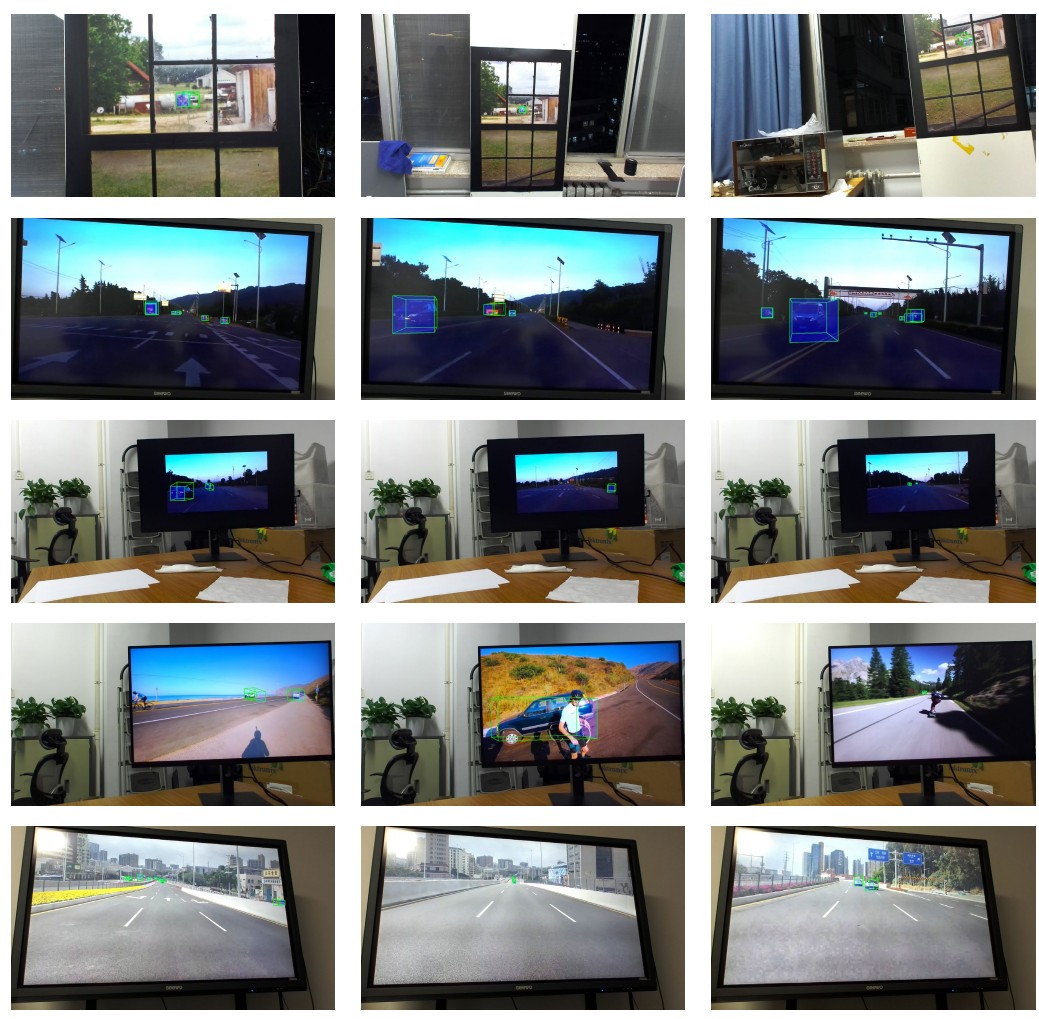

Figure 15: The visualization of 3D detection on real data of 3D-Visual-Illusion dataset.

