# OpenReview forum: "3D Visual Illusion Depth Estimation"
_NeurIPS.cc/2025/Conference — NeurIPS 2025 poster_

### Official Review · Reviewer_hfin · 2025-06-28

**Clarity:** 2
**Significance:** 2
**Originality:** 2
**Rating:** 4
**Confidence:** 2

**Summary:**

This paper addresses the vulnerability of depth estimation models to 3D visual illusions, which manipulate 2D scenes to appear 3D. The authors introduce the 3D-Visual-Illusion dataset, a large-scale benchmark (~3k scenes, 200k images) spanning five illusion types (inpainting, picture, replay, holography, mirror) across virtual and real-world environments. They propose a VLM-driven monocular-stereo fusion model that leverages vision-language models (VLMs) to adaptively fuse monocular depth and binocular disparity cues.

**Questions:**

1.Generalization to Unseen Illusions: The dataset focuses on five illusion types. How does the model handle novel adversarial patterns (e.g., dynamic lighting illusions)? Provide cross-dataset validation (e.g., KITTI-360’s reflective surfaces).
2.Automation in Data Generation: Manual SAM2 masking (Sec. 3.1) is labor-intensive. Can unsupervised segmentation (e.g., DINOv2) reduce human effort?

**Ethical Concerns:**

["NO or VERY MINOR ethics concerns only"]

**Final Justification:**

The author's response has some value, I decided to keep the score

**Limitations:**

-

**Paper Formatting Concerns:**

-

**Quality:**

2

**Strengths And Weaknesses:**

Strengths:
1.First systematic study of 3D illusions’ impact on depth estimation, with actionable insights for safety-critical systems.
2.Novel use of VLMs for confidence-based fusion, distinct from prior monocular-stereo hybrids.

Weaknesses:
1.Ambiguities in virtual data generation (e.g., manual SAM2 masking effort, Sec. 3.1).
2.Insufficient details on VLM fine-tuning (LoRA vs. full-parameter training).

---

> ### Author Rebuttal · Authors · 2025-07-31
>
> ## Response to Reviewer hfin
>
> We sincerely thank the reviewer for the insightful comments and constructive suggestions. Below, we provide detailed responses to each of the concerns.
>
> ### Ambiguities in Virtual Data Generation
>
> Due to the lack of established methods and annotated datasets for detecting 3D visual illusions, we currently rely on manual identification of illusion regions. Prior detection or segmentation methods are easily misled by our 3D visual illusion data. Additionally, the complex content, often involving multiple overlapping elements within a single illusion, requires multiple extensive manual interactions using SAM2 to fully annotate each single illusion region.
>
> ---
>
> ### Insufficient Details on VLM Fine-Tuning
>
> We apply LoRA to fine-tune the last layer of QwenVL2-7B and the QV projection layer of FLUX, as detailed in lines 100–116 of the supplemental materials.
>
> ---
>
> ### Generalization to Unseen Illusions
>
> This is an interesting and challenging suggestion.
>
> 1. Since VLMs often struggle with semantically ambiguous scenes, such as inpainting or printed illusions, we currently restrict their use to identifying *mirror illusions*, which are especially problematic for stereo matching. As long as a novel illusion does not deceive the VLM or both the monocular and stereo models, our approach can effectively handle it. However, when encountering a composition of different illusion types, our method may fail.
>
> 2. Our experiments on **Booster** represent a cross-dataset validation setting. Our model is fine-tuned on our dataset and evaluated directly on the Booster training set. Results indicate that our method achieves over **10 points of improvement** in the `bad x` metric.
>
> ---
>
> ### Automation in Data Generation
>
> This is a promising direction and one we are actively exploring. However, given the near absence of existing datasets for 3D visual illusions, current general-purpose detection and segmentation tools (e.g., SAM, DINOv2) fail to perform reliably. In fact, even VLMs are often fooled by these illusions, and humans ourselves can be misled by complex illusions, as illustrated in Figure 1 of both the main text and the supplemental materials.
>
> Thus, while unsupervised learning alone may currently be insufficient, collecting more diverse and representative data is a practical step forward. Developing automated detection methods for 3D visual illusions remains an exciting direction for future research.

---

> > ### Comment · Reviewer_hfin · 2025-08-06
> >
> > Thanks for your response. I think the main concerns of mine are solved.

---

### Official Review · Reviewer_bw6S · 2025-06-28

**Clarity:** 2
**Significance:** 2
**Originality:** 2
**Rating:** 4
**Confidence:** 4

**Summary:**

The paper presents a 2D depth estimation dataset that particularly consists of 3D illusion examples. To address the 3D illusion issue, the proposed method presents fusing monocular and multi-view depths based on the confidence map learned by VLM based pre-trained network.  The model shows similar performance to the existing stereo methods on proposed 3D illusion dataset and miss qualitative comparisons with them, undermining the importance of VLM-based monocular depth fusion all together. Also, the paper miss comparisons with the related monocular multi-view depth fusion based methods and generalization evaluation making it difficult to consider the method robustness.

**Questions:**

•	Both the disparity maps in the figure4 look same, does that mean if there are no visual illusions the final disparity map does not improve?
•	The reason for utilizing specific stereo GRU model along with the architecture details missing
•	Can the proposed method show improvement in the performance if utilized current SOTA stereo based methods for the first branch?
•	In Figure 4 the color of image and text embeddings in all the stages should be matched as the current one sends confusing information
•	Are existing stereo-based methods able to handle 3D illusion issues as shown by the quantitative comparisons?

**Ethical Concerns:**

["NO or VERY MINOR ethics concerns only"]

**Final Justification:**

I have updated the rating based on the reviews and rebuttal

**Limitations:**

No.
•	The paper should address the limitation of VLM networks that influence performance degradation.
•	The paper should present scenarios where the method might fail.

**Quality:**

2

**Strengths And Weaknesses:**

Strengths:
•	The paper introduced a new dataset to address the 3D illusion issue existing in 2D images.
•	Shows benchmark performance on dataset consisting of transparent/specular surface.

Weakness:
•	The method presents limited novelty in terms of multi-view depth enhancement based on monocular depth fusion, as there exists methods like “Multi-View Depth Estimation by Fusing Single-View Depth Probability” (CVPR 2022), “Learning to Fuse Monocular and Multi-view Cues for Multi-frame Depth Estimation in Dynamic Scenes” (CVPR 2023), “Multi-view Reconstruction via SfM-guided Monocular Depth Estimation” (CVPR 2025) that fuse/utilize monocular depths to enhance multi-view depths estimations.
•	Missing comparison with the related monocular multi-view depth fusion-based methods.
•	The proposed method shows similar performance to the existing stereo-based methods for the presented 3D illusion dataset undermining the requirement for VLM -based monocular depth fusion.
•	Missing zero shot performance evaluation that limits the ability to understand robustness of the approach.
•	The method misses qualitative comparisons with the stereo-based methods on the proposed 3D illusion dataset.

---

> ### Author Rebuttal · Authors · 2025-07-31
>
> ## Response to Reviewer bw6S
>
> ### Limited Novelty
>
> Our key contributions include:
>
> - Identification of new and significant 3D visual illusion challenges to diferent 3D foundation model.
> - Construction of a large-scale dataset featuring diverse simulated and real-world data.
> - Introduction of an effective baseline for addressing these challenges.
>
> As mentioned in Lines 5, 8, and 38 of the main text, our focus is on **monocular and binocular depth estimation**, not multi-view methods. We provide a comparison with VGGT in Tables 1 and 2, which leverages richer monocular information and demonstrates significantly better performance than classical multi-view methods.
>
> Your referenced methods fuse monocular priors into volume, which introduces a strong monocular bias. This early fusion makes the matching process vulnerable to monocular-related illusions such as inpainting, picture, replay, and holography illusions.
>
> ---
>
> ### No Need for VLM-based Monocular Depth Fusion
>
> The comparable performance of our model to stereo-based methods on the 3D-Visual-Illusion dataset arises because its test set comprises real-world data, which **lacks mirror illusions**. Since stereo-based methods are more sensitive to such illusions, we evaluate generalization on the **Booster dataset**, which includes challenging mirror-like regions.
>
> As shown in Table 2, our method **improves the `bad x` metric by over 10 points** in transparent regions, clearly outperforming stereo-based alternatives. Further validation through the ablation study in Table 3 confirms the **effectiveness of the VLM-based monocular depth fusion** approach.
>
> ---
>
> ### Missing Zero-shot Performance Evaluation
>
> The performance shown in Table 1 reflects a generalziation evaluation of our model from virtual data to real-world data. Table 2 further demonstrates zero-shot generalization on the Booster dataset, as described in Lines 235–236 of the main text.
>
> ---
>
> ### Improvements in Non-Visual-Illusion Regions
>
> Contrary to concerns, our model also exhibits **3–5 point improvements in the `bad x` metric** on non-transparent (non-visual-illusion) regions, as presented in Table 2. This confirms that the proposed method not only excels in illusion scenarios but also generalizes well to standard depth estimation settings.
>
>
> ### Utilization of Specific Stereo GRU Model
>
> We use **RAFT-Stereo** as it is a classic and well-known stereo baseline. To investigate the effect of different GRU architectures on fusion performance, we conduct comparisons in Table 3. A detailed explanation of these experiments can be found in Lines 265–272 of the main text.
>
> ---
>
> ### Any Improvement by Using SOTA Stereo-Based Methods?
>
> Using additional SOTA stereo-based methods is indeed a valuable suggestion. However, due to current time constraints, it is not feasible at this stage. Nevertheless, our existing experimental results already validate the **effectiveness of the proposed method**. Incorporating more SOTA stereo methods remains a **promising direction for future work**.
>
> ---
>
> ### Color of Image and Text Embedding in Figure 4
>
> Thank you for your suggestion. We will **change the color of the second text embedding to yellow** in Figure 4 to improve visual clarity and distinguishability.
>
> ---
>
> ### Performance of Existing Stereo-Based Methods on 3D Visual Illusions
>
> Current stereo-based methods continue to struggle with **mirror illusions**, particularly those involving **near mirror surfaces**. What's more, without appropriate fusion strategies, **monocular bias may be injected into the stereo matching process**, leading to degraded performance under visual illusion scenarios.

---

### Official Review · Reviewer_c4xs · 2025-06-28

**Clarity:** 1
**Significance:** 1
**Originality:** 1
**Rating:** 4
**Confidence:** 4

**Summary:**

This paper addresses a really critical, yet under-explored, issue in computer vision: how 3D visual illusions fool machine perception, specifically depth estimation systems. The authors define these illusions as 2D manipulations that simulate 3D spatial relationships, which are known to deceive humans. They reveal that state-of-the-art monocular and binocular depth estimation methods are "seriously fooled" by these illusions, leading to erroneous depth perception. This has significant implications, especially for "safety-critical risks in AR/VR and robotics".

Their main contributions are twofold:
- They introduce a novel, large-scale benchmark dataset called the "3D-Visual-Illusion dataset," comprising "almost 3k scenes and 200k images". This dataset includes five types of illusions: inpainting, picture, replay, holography, and mirror illusions, collected from both virtual and real-world sources.
- They propose a robust depth estimation framework that leverages "commonsense knowledge from a vision-language model (VLM)" to adaptively select reliable depth information from both binocular disparity and monocular depth predictions.

Their experiments show that existing SOTA methods are indeed "all fooled by various 3D visual illusions," while their VLM-driven fusion method achieves "SOTA performance," particularly improving depth estimation in "illusion-affected regions". This highlights a crucial step towards more robust perception systems.

**Questions:**

1. Typo in Ln. 50, Stereo Matching
2. Regarding Model Complexity and Practicality (Weakness #1 & #5b):
The proposed system is a powerful but complex ensemble. For a complete picture of its practicality, could you please:
    - Report the model's inference metrics, including latency (ms/frame), throughput (FPS), and computational requirements (e.g., FLOPs, GPU memory), benchmarked on a specified hardware platform?
3. Regarding the Fairness of Baseline Comparisons (Weakness #2):
    - The paper's central claim rests on the superiority of its fusion architecture, but the current comparisons are structurally asymmetric. To provide a more controlled and compelling evaluation, we ask for an additional experiment:
    - Could you please provide results for the leading baseline models (e.g., DepthAnything v2) after they have also been fine-tuned on the 3D-Visual-Illusion training set? This "apples-to-apples" comparison is crucial for isolating the architectural contribution from the benefits of ensemble complexity and specialized training.
    - Could provide the parameter number comparison with baseline models.

4. Regarding Generalization to Standard Scenes (Weakness #3):
The model shows impressive performance on the target illusion dataset. However, its performance on standard, non-illusory benchmarks is not reported, leaving its broader utility unclear.
   - Could you provide evaluation results on standard benchmarks such as KITTI and NYU Depth V2? This is necessary to assess whether the VLM fusion mechanism introduces any negative interference or performance degradation on more common scenes where traditional methods typically excel.

5. Regarding the VLM's Reasoning Process (Weakness #4):
The VLM's role as a "semantic arbiter" is innovative but largely opaque. To better understand its contribution and failure modes, we request more qualitative analysis.
    - Could you include visualizations of the VLM-generated confidence maps alongside illustrative examples? Such as cases where the VLM successfully identifies an illusion, as well as failure cases where the VLM is also deceived or makes a mistake on a simple, non-illusory scene.
    - Since this is a complex system leveraging lots of prior works, instead of outperforming other models, an extensive ablation study on this system is more important. Could you provide details on how the binocular, monocular, and VLM model is selected, as well as ablations on other model candidates.
    - To better understand the potential failure modes of the VLM's 'common sense,' could the authors comment on its behavior with semantically ambiguous objects? For instance, how does the model classify objects that mimic a property without possessing it, such as people wearing lensless eyeglasses.

6. Regarding Experimental Reproducibility (Weakness #5a):
For the benefit of clarity and reproducibility, we require a more precise description of the baseline setup.

    - In Table 1, please explicitly state for each baseline method whether the reported metrics were derived from a pre-trained checkpoint applied in a zero-shot manner or from a model that was fine-tuned by your team. If using pre-trained models, please specify the exact checkpoints or dataset used.

**Ethical Concerns:**

["NO or VERY MINOR ethics concerns only"]

**Final Justification:**

The authors' new experiments and detailed clarifications in their rebuttal have directly addressed my most critical concerns. They have now:

1. Provided a methodologically sound comparison (Fine-tuned Ours vs. Fine-tuned RAFT-Stereo) that validates their core architectural contribution.

2. Delivered a compelling new experiment with Marigold and offered a plausible technical explanation for the convergence failures of other foundation models.

This effort has fundamentally increased the quality and rigor of the paper. For these reasons, I have raised my score to **Borderline Accept**.

My recommendation of borderline accept is conditional on the authors incorporating the following crucial clarifications and discussions into the final camera-ready version:

* **1. Clarify the Contradictory RAFT-Stereo Result:** The finding that fine-tuning RAFT-Stereo degrades its performance is a surprising and significant claim. The final paper should include the insightful explanation provided in the rebuttal. However, to present this as a general phenomenon, **it is essential that the claim be substantiated** with results from at least one other distinct stereo model showing a similar degradation.

* **2. Be Transparent About VLM Reliability:** The paper's design hinges on the VLM's ability to act as a "mirror detector." The final version should transparently state that a full quantitative analysis of this component was not performed. It should also include the detailed theoretical justification for why the VLM is effective on mirrors but is likely "fooled by other illusion types," even if a full experiment was not conducted.

* **3. Acknowledge the Non-Standard Evaluation Protocol:** Regarding the Middlebury comparison, I acknowledge the authors' stated rationale for using an unrestricted disparity range and your confirmation that all methods were evaluated under the same conditions for internal consistency. However, this choice represents a significant deviation from standard practice, as the cited baseline methods are both architecturally designed for and traditionally evaluated under a restricted maximum disparity. While I am raising my score despite this ambiguity, I strongly recommend that the paper explicitly acknowledge this deviation from standard protocol and discuss how it may affect a direct comparison of your results with those reported in prior work.

* **4. Implement Promised Textual Revisions:** As promised in the rebuttal, the text and table captions throughout the paper should be revised to be unambiguous about which models were fine-tuned for which experiment. This is essential for reader comprehension.

By incorporating these points, the authors will have fully addressed the main issues raised during the review process.

**Limitations:**

Yes

**Quality:**

2

**Strengths And Weaknesses:**

## Strengths
1. **Novel and Important Problem Formulation:** The paper identifies and addresses a critical failure mode in modern depth estimation systems—their vulnerability to 3D visual illusions. This has significant implications for robust perception in real-world applications.

2. **Valuable Dataset Contribution:** The introduction of the `3D-Visual-Illusion` dataset is a major contribution. It provides the community with a standardized benchmark for evaluating model robustness against these specific, challenging scenarios.

3. **Innovative Use of Vision-Language Models (VLMs):** The proposed architecture is forward-thinking. Using a VLM not for direct prediction but as a "common sense" arbiter to fuse geometric cues is a novel approach to integrating semantic understanding into a low-level vision task.

4. **Strong Empirical Results:** The paper demonstrates that its proposed system significantly outperforms existing state-of-the-art models on the new benchmark, validating both the severity of the problem and the effectiveness of their proposed solution.

## Weaknesses
1. **High Model Complexity and Computational Cost:** The system is a heavyweight ensemble of a stereo network, a monocular foundation model, and a large VLM. This makes it computationally expensive and likely slow, posing a significant barrier to practical, real-time deployment.

2. **Potential Unfairness in Baseline Comparisons:**
    - The evaluation could be more rigorous. While the paper shows its fine-tuned model beating zero-shot baselines, it lacks a clear "apples-to-apples" comparison where the baseline models are also fine-tuned on the new dataset. This leaves the source of the performance gain (architecture vs. specialized data) somewhat ambiguous.
    - The evaluation's comparison of a multi-component system against a single module like DepthAnything v2 is structurally imbalanced. The superior performance of such an ensemble is an expected outcome and does not, by itself, provide sufficient evidence for the effectiveness of the proposed fusion architecture over a simpler baseline.

3. **Dependence on a Closed-Domain Dataset:** The model's performance is proven on a specialized dataset of illusions. Its generalization and performance on standard, non-illusory scenes are not fully explored, raising questions about potential performance degradation in mundane scenarios.

4. **Limited Ablation on VLM's Reasoning:** While the VLM is central to the method, its internal decision-making process remains a "black box." A deeper qualitative and quantitative analysis of *why* and *when* the VLM's confidence map makes certain decisions would strengthen the paper's conclusions.

5. **Lack of Clarity**:
    - This paper mentioned several comparisons to baseline SOTA methods. In Table 1, it is compared to DA v2 (metric). Is it also fine-tuned on the 3D Illusion dataset, or which checkpoint did you use?
    - The computational resource for training is listed in supplementary, however, the resource for inference and the inference time/speed isn't stated. It is especially important with such a large system.

---

> ### Author Rebuttal · Authors · 2025-07-31
>
> ## Response to Reviewer c4xs
>
> We appreciate the reviewer’s thoughtful feedback on our model computation, dataset generalization, evaluation fairness, and the role of VLM driven fusion. Below, we address each concern with detailed explanations, additional analyses, and planned revisions.
>
> ---
>
> ### High Model Complexity and Computational Cost
>
> We acknowledge that the current system exhibits high complexity due to the integration of monocular, binocular, and vision-language models. However, we do not claim to have perfectly solved the 3D visual illusion challenge. Rather, our primary contribution lies in systematically investigating **where** and **why** different model types fail, a question of growing importance for downstream applications such as embodied AI, VLM, and automous driving, especially in light of the rapid development of 3D foundation models.
>
> Based on our observations and analyses, we propose a reference baseline that serves as a conceptual and empirical guide to mitigate the impact of this challenge. While we anticipate that future methods will definitely achieve higher efficiency and better performance, we believe that **at this stage, raising awareness of the 3D visual illusion challenge and identifying failure modes across different model paradigms is a more urgent and foundational task**.
>
> To address concerns regarding computational cost, we will include detailed inference metrics (runtime and memory consumption) in the revised experimental section. All results were obtained on a single NVIDIA H100 GPU with input resolution of $1920 \times 1080$:
>
> | Model             | Memory Usage | Inference Time (per iteration) |
> |------------------|--------------|--------------------------------|
> | RAFT-Stereo       | 5610 MB      | 0.87 s/it                      |
> | DepthAnything V2  | 3584 MB      | 0.18 s/it                      |
> | Ours              | 53959 MB     | 4.77 s/it                      |
>
>
> ---
>
> ### Potential Unfairness in Baseline Comparisons
>
> We ensure fair and consistent comparisons across baselines. The models in the ablation study presented in Table 3 of the main text are all fine-tuned on the 3D-Visual-Illusion dataset and evaluated on the Booster training set, as noted in Lines 235–237. The first row of Table 3 represents our **stereo-only baseline model**, which significantly underperforms compared to our full model. Its performance is even worse than RAFT-Stereo without fine-tuning, as shown in Table 2, highlighting the limitations of prior architectures on large-scale illusion data.
>
> Regarding the **monocular-only model**, DepthAnything V2 failed to converge on the 3D-Visual-Illusion dataset. Despite extensive attempts, including data augmentation (e.g., reducing non-illusion regions), multi-stage training (e.g., pre-training on partial regions), and modified loss functions (e.g., weighted loss for illusion areas), the model consistently crashed during training and produced noisy, unusable outputs. Consequently, we chose not to report meaningless results in Table 3. We will include this explanation in the revised manuscript.
>
> For **fusion effectiveness**, we evaluated multiple fusion strategies and demonstrated that our VLM-driven fusion yields more than a 10-point improvement in the `bad x` metric, confirming its effectiveness.
>
> ---
>
> ### Dependence on a Closed-Domain Dataset
>
> We appreciate the reviewer’s emphasis on evaluating generalization beyond the proposed dataset.
>
> As shown in **Table 2 of the main text**, we have already reported the **zero-shot performance** of our model on the **non-transparent regions** of the Booster dataset, which represent mundane, non-illusory scenes. Our model consistently outperforms all baselines across all metrics.
>
>
> To further minimize the potential influence of adjacent illusory regions on nearby non-illusion regions in Booster, we additionally conduct a zero-shot generalization evaluation on the well-known **Middlebury dataset**, where scenes are predominantly free from any 3D visual illusions. We compare our model with several SOTA stereo-based approaches using **metric disparity space** over the **entire image**, without restricting the maximum disparity range.
>
> As shown in the table below, our model does **not degrade** in performance on these mundane scenes. On the contrary, it achieves **improvements**, particularly in terms of EPE:
>
> | Metric       | Selective-RAFT | Selective-IGEV | MochaStereo | StereoAnything | RAFT-Stereo | Ours     |
> |--------------|----------------|----------------|-------------|----------------|-------------|----------|
> | EPE (full)   | 2.34           | 2.59           | 2.66        | 2.89           | 1.92        | **1.50**  |
> | Bad-2 (full) | 12.04          | 11.79          | 10.18       | 11.93          | 12.60       | **11.79** |
>
> These results confirm that our model maintains or improves performance in standard, non-illusory environments, supporting its generalizability and robustness across domains.
>
>
> ---
>
> ### Limited Ablation on VLM's Reasoning
>
> We thank the reviewer for highlighting the need for a deeper analysis of the VLM’s reasoning capabilities.
>
> In response, we provide a quantitative evaluation of the VLM’s confidence by comparing the predicted confidence maps with the disparity error maps, as defined in **Equation 10** of the supplemental material. This analysis is conducted on the **Booster dataset** under a **zero-shot generalization** setting. The results show that the error rate of our confidence estimation is approximately **20%**, demonstrating a strong generalization ability, even in previously unseen illusory scenes.
>
>
> ---
>
> ### Clarity of Model Checkpoint in Table 1
>
> We appreciate the reviewer’s request for clarification on the comparison setup in Table 1.
>
> The methods listed in **Table 1** are all pretrained models evaluated without any fine-tuning on our dataset. All model weights are obtained from their respective official repositories. Specifically, the checkpoint for DepthAnything V2 (metric) is taken from the official GitHub repository under the *Indoor* setting. We will include detailed model checkpoint sources and configurations in the revised version for full transparency and reproducibility.
>
>
>
> ---
>
> ### More Ablation Study on Model Candidates
>
> We acknowledge the reviewer’s interest in additional ablation studies across different model candidates. However, as noted in Line 116 of the supplemental material, the training time for each model variant is nearly **20 days**, making it impractical to conduct further large-scale ablations within the current project scope. We hope the following clarification is satisfactory.
>
> In our study, we carefully selected well-established and widely adopted architectures for both monocular and binocular baselines. For the vision-language component, we employed the most powerful open-source model available from **Qwen** at the time of experimentation.
>
>
> ---
>
> ### Semantically Ambiguous Objects
>
> The suggestion regarding semantically ambiguous failure cases is quite interesting. And we will add more failure case analyses in the supplemental material.
>
> Notably, VLM models struggle in **semantically ambiguous scenes**, such as inpainting or printed illusions. These illusions can mislead even VLMs. Therefore, we currently restrict the use of VLMs to identifying **mirror illusions**, which are particularly problematic for stereo matching.
>
> The solution of our model currently focuses on real-world illusions with **potential risks in downstream applications**, e.g., a printed road image on a wall. We do not explore highly artistic or contrived illusions (e.g., simulated 3D windows behind iron frame), which lie outside the current scope but are an important direction for future exploration.

---

> > ### Comment · Reviewer_c4xs · 2025-08-06
> >
> > Thank you for the detailed and comprehensive rebuttal. The new experiments and clarifications have addressed several of my initial concerns. However, some significant questions regarding the evaluation's fairness and the method's true scope remain.
> >
> > ### **1. Regarding the Ambiguity of Comparisons in Tables 1 & 2**
> >
> > This remains my most critical point of concern. The rebuttal clarified the setup for the ablation study (Table 3), but it seems to have misinterpreted the original question, which concerned the baseline comparisons in Table 1, not Table 3. The ambiguity in this initial comparison persists.
> > * First, for absolute clarity on Table 1: Could you please explicitly confirm that the row for "Ours" was also evaluated in a zero-shot manner, without any fine-tuning on the 3D-Visual-Illusion dataset? If so, we strongly recommend revising the general statement in Ln. 235 (about fine-tuning "the models") to specify that this procedure applies only to your final model presented in later tables, as the current wording is a source of significant confusion.
> >
> > * Second, regarding the performance on illusion regions in Table 2: If your model's superior performance comes after being fine-tuned on the 3D-Visual-Illusion dataset, how can you definitively attribute this success to the architecture's design, rather than the confounding benefit of its specialized training data—an advantage the baselines were not given?
> >
> > ### **2. Regarding the True Generality and Contribution of the VLM**
> >
> > Your rebuttal reveals a crucial detail that seems to significantly narrow the scope of your method's contribution.
> >
> > * You state that "**we currently restrict the use of VLMs to identifying mirror illusions.**" This suggests the VLM is not a general semantic arbiter but a specialized "mirror detector." Should the paper's main claims be revised to more accurately reflect that the current implementation is a specialized solution for mirrors, rather than a general framework for all 3D illusions?
> >
> > ### **3. Regarding the New Experiment on Middlebury**
> > Thank you for providing additional results. To ensure this new comparison is sound, I would like to ask for one more clarification.
> > * Could you please clarify the rationale for evaluating your model "without restricting the maximum disparity range"? As most state-of-the-art methods report results under a restricted range.
> >
> > ### **4. Regarding the "Failure to Converge" Claim**
> >
> > The claim that a SOTA model like DepthAnything V2 fails to converge is a very strong one and a key part of your defense.
> > * To strengthen this claim, have you confirmed whether this convergence failure is a general issue for other leading monocular foundation models, or if it is specific to DepthAnything V2? If the failure is unique to DepthAnything V2, then the selection of this particular model requires stronger justification.
> >
> > Addressing these final points will resolve the remaining ambiguities and allow for a more accurate assessment of your work's important contributions.

---

> > > ### Author Response · Authors · 2025-08-07
> > > **Regarding the Ambiguity of Comparisons in Tables 1 & 2**
> > >
> > > We sincerely thank you for your thoughtful and valuable feedback. We truly appreciate the time and effort you spent reviewing our work. Below, we provide point-by-point responses to your comments and concerns.
> > >
> > >
> > > ### **1. Regarding the Ambiguity of Comparisons in Tables 1 & 2**
> > >
> > > - For Tables 1 and 2, **only our method is fine-tuned** on the 3D-Visual-Illusion dataset. All other methods, including the baselines (RAFTStereo and DepthAnything V2), use **pretrained models from official repositories without any fine-tuning** on our dataset.
> > >
> > >   The purpose of these two tables is to demonstrate the **significant impact of 3D visual illusions** on various SOTA methods, to **raise awareness of the challenges** posed by such illusions, and to **identifying failure modes across different model paradigms**.
> > >
> > >   To avoid confusion, we will **clarify in the captions of Tables 1 and 2** that only our method is fine-tuned, while others remain at their official pretrained settings. We also notice that **Lines 257–258** in the main text may lead to this misunderstanding, and we will revise them to focus specifically on **explaining the impact of mirror illusions across model paradigms**, rather than only highlight the performance of our method.
> > >
> > >
> > > - This is a great and valuable question. We also notice that the performance improvements may stem from **either the dataset, the architecture, or both**. To address this, we include a direct comparison to the **RAFTStereo baseline**:
> > >
> > >   The `RAFTStereo` row under the `Trans` column in **Table 2** shows results of `9.55, 67.84, 59.43, 47.46`, which are obtained from the **official pretrained model without fine-tuning**.
> > >
> > >   In contrast, the **same RAFTStereo architecture**, when **fine-tuned on our 3D-Visual-Illusion dataset**, achieves
> > >   `15.11, 80.38, 72.35, 61.32` in **Table 3 (first row)**. They are evaluated under the same setting on **transparent regions of Booster**.
> > >
> > >   The only difference between the two is whether fine-tuning on our dataset was applied. This comparison **highlights the limitation of existing architectures when exposed to large-scale 3D visual illusion data**.
> > >
> > >   Furthermore, Table 3 explores various fusion architectures, demonstrating their increased robustness when trained on 3D illusion data. We will include the above explanation in the experimental section to better emphasize the **critical role of large-scale illusion data in model training** and the **value of architectural enhancements**.

---

> > > > ### Author Response · Authors · 2025-08-07
> > > > **Regarding the True Generality and Contribution of the VLM**
> > > >
> > > > ### **2. Regarding the True Generality and Contribution of the VLM**
> > > >
> > > > We acknowledge that there are truly many limitations of our method when applied to such a completely new and challenging problem.
> > > >
> > > > (1) Yes, the **VLM is not a general-purpose semantic arbiter**, as stated in lines 204–206 of the main text. We will emphasize this point more clearly in the Method section.
> > > >
> > > > (2) Our current model is **not a mirror-specific solution**, but is designed to handle five types of 3D visual illusions: **inpainting, picture, replay, holography, and mirrors**. We clarify this by first explaining what we mean by the statement: *“we currently restrict the use of VLMs to identifying mirror illusions.”* This is a **design decision** informed by two key observations:
> > > >
> > > >
> > > > - First, **different types of illusions affect different depth estimation paradigms**. Monocular methods are easily misled by **inpainting, picture, replay, and holography**, which simulate depth cues through RGB textures. In contrast, **stereo-based methods** typically fail in the presence of **mirror illusions**, where pixel-level correspondence becomes ambiguous. Therefore, we only need to distinguish **mirror** versus **non-mirror** illusions, and fuse stereo and monocular depth accordingly to handle depth estimation on all five illusions.
> > > >
> > > > - Second, **VLMs show reliable performance in detecting mirror illusions**, as demonstrated in Figure 5 of the supplementary materials. However, they are often **fooled by the other illusion types**. For this reason, we currently apply the VLM **only to detect mirrors**, while the overall model integrates stereo and monocular depth predictions to handle all five illusion types.
> > > >
> > > > We further explain how these illusions affect each depth paradigm:
> > > >
> > > > - **Monocular depth estimation** relies on visual cues such as **shape, perspective, shadow, and defocus**, which are learned from RGB textures. When these cues are artificially simulated on flat surfaces, like in **inpainting, printed images, replays, or holograms**, the monocular model is easily **misled**, resulting in incorrect depth estimation. Interestingly, **mirror illusions** can be handled correctly due to scene-level reasoning abilities learned from large-scale datasets.
> > > >
> > > > - **Stereo-based depth estimation** depends on **pixel-wise matching** between image pairs. In mirror scenes, both the real surface and its reflected background are projected onto the same regions, causing **ambiguous correspondences**. Because the background dominates the RGB signal, the stereo model often **ignores the actual mirror surface**, resulting in incorrect depth. However, for **inpainting, picture, replay, and holography illusions**, stereo matching remains effective due to consistent texture between views, even if the resulting geometry is illusory.
> > > >
> > > > We hope this clarifies both the **scope and limitations of the VLM**, and the **generalization capability of the entire model** across diverse 3D visual illusions.

---

> > > > > ### Author Response · Authors · 2025-08-07
> > > > > **Regarding the New Experiment on Middlebury**
> > > > >
> > > > > ### **3. Regarding the New Experiment on Middlebury**
> > > > >
> > > > > We do not restrict the maximum disparity range in our evaluation, as our goal is to assess performance **across the entire image**, rather than focusing on only a portion of objects within the scene.
> > > > >
> > > > > In practice, real-world (or "in-the-wild") images vary in resolution and scene content, leading to **different maximum disparity ranges** depending on the image. To ensure comprehensive evaluation, especially for **near objects** that may exhibit large disparities, it is preferable to **avoid enforcing a fixed disparity range** across all images.

---

> > > > > > ### Author Response · Authors · 2025-08-07
> > > > > > **Regarding the "Failure to Converge" Claim**
> > > > > >
> > > > > > ### **4. Regarding the "Failure to Converge" Claim**
> > > > > >
> > > > > > We claimed "DepthAnything V2 failed to converge on the 3D-Visual-Illusion dataset". In fact, we initially questioned whether there might be issues with our own training code. After carefuly checking, actually we only need to check several modified lines of official code, there is no problem. And the training on RAFT-stereo and our models went well, showing the data is also OK.
> > > > > >
> > > > > > During our experiments, we directly used **DepthAnything V2** as the baseline for monocular depth estimation, since it was the **most powerful model with clear open-source code** during our work period. We did not check this phenomenon with other monocular models.
> > > > > >
> > > > > > To further investigate this phenomenon, we are currently conducting additional experiments using **Marigold**, a diffusion-based framework that is architecturally very different from DepthAnything V2. These results will help us verify whether the same convergence issues occur across different monocular paradigms. Please wait for at least one day, we will provide as least the intermediate training results.
> > > > > >
> > > > > > We also offer a explanation for this convergence problem. As discussed earlier in our response to *“2. Regarding the True Generality and Contribution of the VLM,”* monocular depth estimation methods learn **fixed mappings from texture to geometry**. In the presence of illusions, such as a **real apple** versus a **printed apple image captured from the same viewpoint**, the textures may be nearly identical. This creates an **inherent ambiguity**: the model cannot reliably decide whether to regress to a 3D object or a flat surface. Such ambiguity can easily **corrupt the learning process**, making convergence difficult or unstable.
> > > > > >
> > > > > > ---
> > > > > >
> > > > > > We hope the above clarification and the forthcoming analysis will help address your concerns effectively. We acknowledge that our current model may not fully handle **unseen types of illusion** or **complex composite illusions**. However, given the novelty and difficulty of this completely new challenge, we kindly ask you to consider our **contributions beyond model performance**, including (1) the **identification of the 3D visual illusion challenge**, (2) the construction of a diverse **dataset** along with a complete pipeline for **web-sourced** and **generative-model-sourced** data, (3) and the **systematic investigation** into how 3D visual illusions affect **different model paradigms**. We believe these efforts provide a valuable foundation for future research in this emerging area, such as 3D reconstruction \& detection \& segmentation, 3D foundation model, VLM, VLA, and embodied AI.

---

> > > > > > > ### Author Response · Authors · 2025-08-08
> > > > > > > **Extended Evaluation and Insights on Monocular Depth Model**
> > > > > > >
> > > > > > > Thanks for waiting. Below are the results of our additional analysis using **Marigold**, along with our observations and interpretation.
> > > > > > >
> > > > > > > |     Region     | Finetuning | EPE ↓  | bad 2 ↓ | bad 3 ↓ | AbsRel ↓ | δ₁ ↑  |
> > > > > > > |----------------|:----------:|--------|---------|---------|----------|-------|
> > > > > > > |   Illusion     |     ❌     | 21.16  |  65.67  |  59.67  |  0.45    | 63.65 |
> > > > > > > |   Illusion     |     ✅     | 13.67  |  74.82  |  55.20  |  0.28    | 71.04 |
> > > > > > > | Non-illusion   |     ❌     |  7.61  |  49.18  |  39.56  |  0.18    | 79.76 |
> > > > > > > | Non-illusion   |     ✅     |  7.10  |  55.63  |  44.09  |  0.16    | 77.44 |
> > > > > > >
> > > > > > > We observe that **almost all metrics improve on illusion regions after fine-tuning**, indicating that Marigold gradually adjusts its predictions on illusion regions toward planar surfaces. However, this improvement comes at the cost of **degraded performance on non-illusion regions**. This contrast further supports our hypothesis: **monocular depth models learn fixed mappings from texture-based cues (e.g., shape, shadow, perspective, defocus) to geometry**, and it is challenging for them to differentiate between **illusion textures** and **real 3D object texture**. We also provide additional analysis from the table. The bad 2 metric worsens in illusion regions after fine-tuning (row 2 vs. row 1), likely because Marigold is not fully overfitting to the illusion data. The limited EPE improvement in non-illusion regions (row 4 vs. row 3) may result from the correction of a few extreme outlier noise points, rather than a consistent gain.
> > > > > > >
> > > > > > > Beyond the quantitative analysis, we also conducted **qualitative visualization**, which will be presented in the supplementary materials. Interestingly, while some illusions with **uncommon objects or low-quality images** become more planar after fine-tuning, **illusions involving daily-life objects and high-quality images** tend to produce more **three-dimensional and sharp depth predictions**, e.g., a large portrait with a coffee cup on the table.
> > > > > > >
> > > > > > > We also revisited the **failure to converge in DepthAnything V2**. We now figure out the failure stems from the training objectives: the **official DepthAnything V2 code is implemented in training on metric depth**, whereas **Marigold provides the codes for training on affine-invariant depth**, which is more robust to **scale variations** present in our generated data. Since our data varies significantly in scale across different samples, metric-depth-based training becomes unstable or ineffective.
> > > > > > >
> > > > > > > Finally, we once again thank the reviewer for this valuable comment. The 3D visual illusion is a **completely new and challenging problem**, and many open directions remain to be explored. Our work aims to bring **awareness to 3D visual illusions** for our community, particularly from the perspective of depth estimation. We contribute by formulating this new challenge, constructing **a diverse dataset** with both real and synthetic data, and the corresponding **generation pipeline** for web-sourced and generative-model-sourced data. We also conduct a **systematic study of how illusions affect different model paradigms** and provide a **reference baseline** for 3D visual illusions. We hope these efforts can provide meaningful insights and a foundation for future research for our community.

---

> > > > > > > ### Comment · Reviewer_c4xs · 2025-08-08
> > > > > > > **4. Regarding the "Failure to Converge" Claim**
> > > > > > >
> > > > > > > Thank you for this transparent and proactive response. Your theoretical explanation for the convergence failure is insightful.
> > > > > > >
> > > > > > > The new experiment you are conducting with Marigold addresses my primary remaining concern. Its outcome is the crucial piece of evidence needed to validate your central claim. Therefore, including these results will be an essential and important addition to your paper.

---

> > > > > > > > ### Author Response · Authors · 2025-08-08
> > > > > > > >
> > > > > > > > Thank you very much for your thoughtful suggestion. We will include the results and detailed analysis in the revised version of our experiments and supplementary materials. We sincerely appreciate your insightful and constructive comments.

---

> > > > > > ### Comment · Reviewer_c4xs · 2025-08-08
> > > > > > **3. Regarding the New Experiment on Middlebury**
> > > > > >
> > > > > > Thank you for clarifying your rationale for using an unrestricted disparity range to better assess performance on "in-the-wild" scenes. I understand the reasoning.
> > > > > >
> > > > > > However, my question was about the **methodological fairness of the comparison** itself. A valid scientific comparison requires that all methods be evaluated under identical conditions.
> > > > > >
> > > > > > To that end, please clarify: were the baseline results you cited in the new Middlebury table originally designed / trained / reported by their authors under the same **unrestricted** disparity setting that you used for your model?

---

> > > > > > > ### Author Response · Authors · 2025-08-08
> > > > > > > **Ensuring Fair Comparison in Middlebury Evaluation**
> > > > > > >
> > > > > > > Yes, we can confirm that all methods listed in the new Middlebury table were evaluated using their **official code and pretrained weights**, and all results were obtained using **our unified evaluation code** to ensure fair and consistent comparison.

---

> > > > > ### Comment · Reviewer_c4xs · 2025-08-08
> > > > > **2. Regarding the True Generality and Contribution of the VLM**
> > > > >
> > > > > Thank you for the detailed explanation. I understand that the VLM was chosen for its **scene-level reasoning** capabilities, which you found to be effective for the specific task of mirror detection, while other illusions are handled by the complementary stereo/mono branches. This is a clear and logical design.
> > > > >
> > > > > However, this design's validity now rests entirely on two key claims you've made about VLM performance. I ask for the specific evidence to substantiate them:
> > > > >
> > > > > 1.  **Regarding the VLM's Reliability on Mirrors:**
> > > > >     You state that "**VLMs show reliable performance in detecting mirror illusions.**" To quantify this claim and assess the robustness of the system's core "switch," could you please provide the specific binary classification metrics (e.g., precision, recall, F1-score) for this task?
> > > > >
> > > > > 2.  **Regarding the VLM's Unreliability on Other Illusions:**
> > > > >     Your justification for limiting the VLM's role is that they are "**fooled by the other illusion types.**" Does the paper contain a systematic evaluation or cite evidence to support this second claim? This is necessary to fully justify the VLM's specialized role in your architecture.
> > > > >
> > > > > Clarifying these two points by providing the quantitative data is essential for validating your design choices.

---

> > > > > > ### Author Response · Authors · 2025-08-08
> > > > > > **On the Reliability and Limitations of VLMs in Illusion Understanding**
> > > > > >
> > > > > > ### 1. Regarding the VLM's Reliability on Mirrors
> > > > > >
> > > > > >   Sorry, we did not conduct a dedicated quantitative analysis focused solely on the VLM. However, prior to starting our work, we conducted a preliminary test on 30 images using SOTA VLMs to assess their capability. We found that **most VLMs were able to correctly identify the presence of a mirror in the image**, consistent with the behavior illustrated in **Figure 5** of the supplementary materials.
> > > > > >
> > > > > >   The most challenging examples involved **movie scenes with carefully designed lighting and shadow deception**, such as those in *Inception*. In these difficult cases, we observed that **VLMs with stronger reasoning capabilities performed noticeably better**.
> > > > > >
> > > > > >   We also experimented with prompting the VLM to generate **segmentation masks** indicating the precise location of the mirror. However, the results were **incomplete and inconsistent**, often capturing only parts of the mirror and falling short of practical utility. That is why we further use **flow matching to derive a confidence map** from the **VLM's latent embeddings**.
> > > > > >
> > > > > >   As discussed in our response to *“Limited Ablation on VLM's Reasoning”*, we conducted an analysis on the **Booster dataset** under a **zero-shot generalization setting**. The results show that the **error rate of our VLM-based confidence estimation is approximately 20%**, indicating **strong generalization**, even in previously unseen and challenging illusion scenarios.
> > > > > >
> > > > > >
> > > > > >
> > > > > > ### 2. Regarding the VLM's Unreliability on Other Illusions
> > > > > >
> > > > > >   We did not conduct a systematic evaluation of VLM performance on inpainting, picture, replay, and holography illusions. Instead, we **sampled 100 images** from the collected videos during the bad frame filtering stage for qualitative testing.
> > > > > >
> > > > > > We found that when the camera view is distant and there are **clear contextual cues**, such as a **picture frame separating the illusion from its surroundings**, the VLM can often correctly identify the illusion, **provided that a precise prompt is given**. However, the **content and appearance of illusions are highly diverse**, making it difficult to construct universally effective prompts.
> > > > > >
> > > > > > As mentioned in Lines 119–120 of the supplemental materials, we experimented with **many prompt variations** to filter out illusion-unrelated data as much as possible. Nonetheless, **a significant amount of unrelated data remained**, requiring **manual filtering**. This process is described in detail in Section 3.1 of the main text.
> > > > > >
> > > > > > In many cases, the illusions were **so effective** that even our human selves were misled. We had to **re-watch entire videos again and again** just to determine whether an illusion was truly present. This process was **extremely challenging, time-consuming, and struggling**, and it gave a critical lesson to us: **VLMs are not reliable for handling these types of illusions**, particularly when the prompt is vague or when visual cues are subtle.
> > > > > >
> > > > > > We agree that **studying visual illusions in the context of VLMs is an interesting and important direction**, but it is unfortunately **beyond the scope of this paper**. We believe this line of investigation **deserves a separate, dedicated study** to systematically explore how much VLMs are influenced by visual illusions.

---

> ### Comment · Reviewer_c4xs · 2025-08-08
> **1. Regarding the Ambiguity of Comparisons in Tables 1 & 2**
>
> Thank you for the clarification. The new information is helpful, but it has revealed a significant contradiction in your results and has not resolved our core concern about fair comparisons.
>
> #### **1. Regarding the Need for a Clear and Fair Comparison**
>
> The comparisons in Tables 1 and 2, as they stand, remain methodologically unsound for making performance claims.
>
> * **On your stated goal of "raising awareness":** Simply placing your fine-tuned model next to zero-shot baselines in Table 1 is confusing, not illuminating. A more effective way to demonstrate the challenge would be to restructure the table to directly compare baseline models' performance on **illusion vs. non-illusion regions**.
> * **On performance attribution:** Because the baseline models in Table 2 were not fine-tuned on your illusion dataset, it is impossible to **attribute your model's superior performance to its architecture**. The advantage more likely stems from the specialized training data, a benefit the baselines were not given.
>
> ---
> #### **2. Regarding the Contradictory RAFT-Stereo Results**
>
> You've argued that fine-tuning on your dataset demonstrates its value. However, the results you provided show the exact opposite for the RAFT-Stereo baseline:
>
> | Model | EPE ↓ | bad2 ↓ | bad3 ↓ | bad5 ↓ |
> | :--- | :-: | :-: | :-: | :-: |
> | **Pretrained** RAFT-Stereo | 9.55 | 67.84 | 59.43 | 47.46 |
> | **Fine-tuned** RAFT-Stereo | 15.11 | 80.38 | 72.35 | 61.32 |
>
> As shown, fine-tuning the baseline on your dataset significantly **degraded** its performance across all metrics (e.g., EPE increased from 9.55 to 15.11).
>
> * Could you please explain this major performance drop? This result seems to suggest that standard architectures like RAFT-Stereo are fundamentally unable to learn effectively from your dataset's distribution. This is an important finding and should be stated more clearly with evidence and experiments.
>
> ---
> I believe that addressing these final points will resolve the remaining ambiguities and allow for a more accurate assessment of your work's important contributions.

---

> > ### Author Response · Authors · 2025-08-08
> > **Clarification on Model Comparisons and Fine-Tuning Behavior**
> >
> > Thanks for your reply.
> >
> > ### 1. Regarding the Need for a Clear and Fair Comparison
> >
> > We agree that the current table layout could be improved. In our revision, we will reorganize the tables to separately present the performance of SOTA zero-shot depth estimation models on **illusion vs. non-illusion regions**, and build another dedicated table for **fine-tuned models**.
> >
> > Generally, your point is right. The consistent improvement on all illusions comes from both our data and model. However, directly fine-tuning SOTA models (e.g., RAFT-Stereo) on our illusion data **without modifying the architecture** actually **degrades performance**, as shown in the table below. This phenomenon highlights an inherent challenge for the completely new 3D-Visual-Illusion problem: **a new architecture or training strategy may be required to properly leverage large-scale illusion data and achieve consistent gains on illusions**.
> >
> > In this context, our method demonstrates strong generalization. After fine-tuning with the same illusion data, our model **consistently improves the results on illusions**, validating the **effectiveness of our architectural design**.
> >
> > | Model                  | EPE ↓ | bad2 ↓ | bad3 ↓ | bad5 ↓ |
> > |------------------------|-------|--------|--------|--------|
> > | Pretrained RAFT-Stereo | 9.55  | 67.84  | 59.43  | 47.46  |
> > | Fine-tuned RAFT-Stereo | 15.11 | 80.38  | 72.35  | 61.32  |
> > | Fine-tuned Ours based on RAFT-Stereo | **7.32**  | **56.77**  | **47.83**  | **36.45**  |
> >
> >
> >
> > ### 2. Regarding the Contradictory RAFT-Stereo Results
> > Your insight is truly impressive. Yes, the results suggest that standard architectures like RAFT-Stereo are fundamentally unable to learn effectively from our dataset. In our prior experience, fine-tuning RAFT-Stereo on **mirror-heavy datasets** can bring slight improvements on the Booster dataset. However, when training on **diverse large-scale illusion data**, this improvement disappears. This is due to the following two main reasons:
> >
> > - **Mutual exclusivity between illusion types**
> >
> >   - **Mirror illusions** require models to leverage **spatial context*, a type of monocular priors, for accurate depth estimation.
> >   - In contrast, **inpainting, picture, replay, and holography illusions** often mislead models precisely because they **manipulate these priors** using deceptive textures.
> >
> >   As a result, the features learned from mirror data can be **undermined** by subsequent training on the other four illusion types. Without **a more robust architecture or specialized training strategy**, RAFT-Stereo tends to **overfit to the dominant illusion types**, leading to degraded performance overall.
> >
> >
> > - **Unbalanced dataset distribution**
> >
> >   Our dataset is built from (1) **4,519 videos with 1.4M frames** for inpainting, picture, replay, and holography illusions (2) and  **234 videos with 2,382 frames** for mirror illusions.
> >
> >   This imbalance causes the model to focus predominantly on the first four illusion types, where the spatial context information is fooled and the model is forced to rely only on matching information. When meeting a mirror illusion, spatial context information becomes more important, and matching will be fooled. As a result, the model learns to **downplay spatial context cues**, degrading its performance on mirror-heavy datasets such as Booster after fine-tuning.
> >
> > ---
> >
> > We hope these clarifications help explain the observed results and further highlight the **unique challenges** of 3D visual illusions as a new research exploration direction.

---

### Official Review · Reviewer_ud34 · 2025-07-03

**Clarity:** 2
**Significance:** 3
**Originality:** 1
**Rating:** 4
**Confidence:** 5

**Summary:**

This paper systematically investigates the degradation of depth estimation performance in the presence of 3D visual illusions. The work's contribution is twofold: (1) the construction of a comprehensive and large-scale “3D-Visual-Illusion” dataset for benchmarking, and (2) the development of a novel monocular-stereo fusion framework guided by a Vision-Language Model (VLM). Experiments on the proposed dataset and the Booster dataset show that the proposed method can achieve advanced depth accuracy.

**Questions:**

Please refer to the major weaknesses.

**Ethical Concerns:**

["NO or VERY MINOR ethics concerns only"]

**Final Justification:**

Thank you for the detailed response. This clarifies most of my questions. I decided to keep the score.

**Limitations:**

1. The quality of the figures should be further improved, Figures 2 to 3.
2. The writing of the paper should be further improved.

**Quality:**

3

**Strengths And Weaknesses:**

Strengths:

S1. They introduce the "3D-Visual-Illusion" dataset, a pioneering, large-scale benchmark specifically designed to test algorithmic robustness in such scenarios.

S2. Most ablation experiments are comprehensive.


Weaknesses:

W1. A significant methodological concern lies in the generation of "ground truth" for the virtual data. The authors initiate the process using a monocular depth estimation model (e.g., DepthAnythingV2), yet the paper's core premise is that these very models are severely deceived by 3D visual illusions. This creates a potential paradox. The subsequent correction process, which heavily relies on manual annotation and a rigid "co-planar assumption" for plane fitting, is problematic. This assumption is often violated in practice (e.g., a picture on a curved surface), and this combination of manual intervention and model-based correction may introduce systematic bias and inaccuracies into the dataset.

W2. A primary weakness of this work is the conceptual gap between the problem it identifies and the solution it proposes. While the paper successfully highlights that 3D visual illusions degrade depth estimation performance, it stops short of a principled analysis of the underlying mechanisms. It does not sufficiently dissect why and how different types of illusions (e.g., pictorial vs. reflective) specifically cause monocular or stereo-based algorithms to fail. This lack of deep analysis leads to a subsequent issue: the connection between the problem and the proposed VLM-driven framework feels tenuous. The paper posits that the VLM provides "common sense" to guide the fusion, but it fails to establish a clear causal link showing how the VLM's reasoning directly counteracts the specific failure modes of visual illusions.

---

> ### Author Rebuttal · Authors · 2025-07-31
>
> ## Response to Reviewer ud34
>
> We appreciate the reviewer’s thoughtful concerns regarding the generation of ground truth for the virtual data and the potential paradox raised. Below, we provide detailed responses to each point.
>
> ---
>
> ### **1. Purpose of Initial Depth Model Use**
>
> We adopt **DepthAnything V2** as our initial depth predictor because it is the **only monocular foundation model** that consistently produces **generalizable and dense depth predictions across diverse scenes**. In contrast, classical multi-view reconstruction methods fail to generate satisfactory dense depth for our **web-sourced monocular videos**, primarily due to the following challenges:
>
> - Presence of **dynamic objects**
> - **Complex lighting** conditions
> - **Fixed or limited camera motion**, which degrades the quality of multi-view reconstruction
>
> As mentioned in Lines 108–111, these characteristics make classical multi-view approaches unsuitable for our setting.
>
> ---
>
> ### **2. Manual Annotation of Illusion Masks**
>
> Since **no existing method or dataset** provides reliable detection of various 3D visual illusions, **manual annotation** is currently **the only viable solution** for creating illusion masks, which are crucial for both **training and evaluation**.
>
> While we agree that developing an **automatic monocular 3D illusion detector** would enhance scalability, such work is **beyond the scope of this paper** and warrants **future investigation**.
>
> ---
>
> ### **3. Co-Planar Assumption and Systematic Bias**
>
> We acknowledge the concern regarding potential **systematic bias** introduced by the co-planar assumption. However, we argue that this effect is limited due to two key reasons:
>
> 1. **Deformation reduces illusion effectiveness**: In real-world scenarios, daily-life illusions lose their illusory effect when displayed on **curved or irregular surfaces**. Moreover, illusions specifically designed for curved surfaces are **extremely rare**, less than **800 images** in our dataset that contains more than **200,000 images**.
>
> 2. **Robustness in binocular disparity learning**: In rare cases where surface is curved and planar fitting is inaccurate, the **pixel-wise matching** mechanism in stereo models **remains robust**, because **learning is minimally affected** as long as matching is correct, no matter the surface is curved or not in zero-shot generalization testing.
>
> ---
>
> ### **4. Link Between Illusions and Specific Algorithms**
>
> We thank the reviewer for this insightful suggestion. The following explanation will be added to the Introduction and Method sections:
>
> #### (1) Monocular Depth Estimation:
> Monocular methods rely on depth cues such as **shape, perspective projection, shadow, and defocus**, which are embedded in RGB textures. These models learn **fixed mappings** from texture to geometry. However, when **visual illusions simulate such texture cues on a 2D surface** (e.g., via inpainting, printed pictures, replays, or holograms), the learned mappings are **misled**, resulting in **incorrect depth predictions**, which manifests as an illusory effect.
>
> #### (2) Stereo-Based Depth Estimation:
> Stereo methods depend on **pixel-wise matching** across views. Reflective or transparent surfaces cause **ambiguous correspondences** as both the real surface (e.g., mirror) and virtual background are projected onto the same pixels. Since most RGB signals come from the distant background, the near mirror surface is often **ignored**, leading to **incorrect depth** and a **visual illusion**.
>
> #### (3) More Analysis:
> - For inpainting, picture, replay, and holography illusions, stereo-based methods will find right and great matching between left and right images according to the rich texture.
> - For mirror illusions, monocular methods will predict correct geometry due to scene understanding abilities learned from large-scale datasets.
>
> ---
>
> ### **5. Connection Between the Problem and the VLM-Based Solution**
>
> As illustrated in **Figure 5 of the supplementary materials**, Vision-Language Models (VLMs) can **reliably recognize mirror illusions** and provide **reasoned judgments**. Motivated by this, we use VLMs with **carefully designed prompts** to provide **commonsense-level reasoning** for detecting mirror illusions.
>
> This enables the generation of a **confidence map**, indicating regions where **binocular-based methods are likely to fail**. The implementation details are discussed in **Lines 202–206** of the main text and **Lines 137–140** of the supplementary materials.
>
> ---
>
> We hope these clarifications effectively address the reviewer’s concerns and help improve the understanding of our work. Thank you again for the valuable feedback.

---

### Decision · Program_Chairs · 2025-09-17

**Decision:**

Accept (poster)

**Comment:**

In the paper the authors systematically investigate the degradation of depth estimation performance in the presence of 3D visual illusions. The authors provide a comprehensive and large-scale “3D-Visual-Illusion” dataset for benchmarking. They also develop a novel monocular-stereo fusion framework guided by a Vision-Language Model (VLM). Experiments on the proposed dataset and the Booster dataset show that the proposed method can achieve advanced depth accuracy.
Strengths of the paper include (i) Interesting and important problem, (ii) New dataset for benchmark, (iii) Innovative Use of Vision-Language Models, (iv) Strong Empirical Results, (v) Solid ablation study.
Weaknesses include
Ground truth - how good is it
Lacking analysis of the core of the problem
High model complexity
Potential Unfairness in Baseline Comparisons
The evaluation's comparison of a multi-component system against a single module like DepthAnything v2 is structurally imbalanced
Lack of Clarity
Somewhat limited novelty
Missing comparison with the related monocular multi-view depth fusion-based methods
Missing zero shot performance evaluation
The authors provide a good rebuttal. Solid discussion between authors and reviewers, after which the reviewers were predominantly positive with grades 4,4,3,4. Although one reviewer rated it as borderline reject, I still recommend accepting the paper.